# Learning 3D Perception from Others' Predictions

**Jinsu Yoo**[1] **Zhenyang Feng**[1] **Tai-Yu Pan**[1] **Yihong Sun**[2] **Cheng Perng Phoo**[2] **Xiangyu Chen**[2]
**Mark Campbell**[2] **Kilian Q Weinberger**[2] **Bharath Hariharan**[2] **Wei-Lun Chao**[1]
[1]The Ohio State University        [2]Cornell University

## Abstract

Accurate 3D object detection in real-world environments requires a huge amount of annotated data with high quality. Acquiring such data is tedious and expensive, and often needs repeated effort when a new sensor is adopted or when the detector is deployed in a new environment. We investigate a new scenario to construct 3D object detectors: *learning from the predictions of a nearby unit that is equipped with an accurate detector.* For example, when a self-driving car enters a new area, it may learn from other traffic participants whose detectors have been optimized for that area. This setting is label-efficient, sensor-agnostic, and communication-efficient: nearby units only need to share the predictions with the ego agent (*e.g.*, car). Naively using the received predictions as ground-truths to train the detector for the ego car, however, leads to inferior performance. We systematically study the problem and identify viewpoint mismatches and mislocalization (due to synchronization and GPS errors) as the main causes, which unavoidably result in false positives, false negatives, and inaccurate pseudo labels. We propose a distance-based curriculum, first learning from closer units with similar viewpoints and subsequently improving the quality of other units' predictions via self-training. We further demonstrate that an effective pseudo label refinement module can be trained with a handful of annotated data, largely reducing the data quantity necessary to train an object detector. We validate our approach on the recently released real-world collaborative driving dataset, using reference cars' predictions as pseudo labels for the ego car. Extensive experiments including several scenarios (*e.g.*, different sensors, detectors, and domains) demonstrate the effectiveness of our approach toward label-efficient learning of 3D perception from other units' predictions.[1]

## 1 Introduction

Accurate detection of mobile objects (*e.g.*, vehicles, humans) in 3D is essential for an intelligent agent (*e.g.*, self-driving car, service robot) to operate safely and reliably (Lang et al., 2019; Shi et al., 2019; Wang et al., 2019; Shi et al., 2020). Constructing such a 3D object detector is never easy — it requires a huge amount of high-quality *labeled* data. Acquiring them is laborious and expensive, and is seldom a once-and-for-all effort. Whenever an agent enters a new environment and encounters new objects, its detector needs adaptation to remain accurate. Whenever a new sensor is adopted (*e.g.*, for energy or space efficiency), the different patterns in sensor data (*e.g.*, LiDAR point cloud style and density) necessitate the detector to be retrained. All these updates to the detector imply yet another round of tedious labeled data acquisition.

*Could we bypass or, at least, reduce the repeated labeling effort?* In this paper, we investigate the scenario in which there are other nearby agents equipped with accurate 3D object detectors (but not necessarily with the same sensor configuration). This scenario is realistic and promising. For example, self-driving taxis (*e.g.*, Waymo, Baidu) or local facilities (*e.g.*, surveillance systems, roadside units) are likely to be equipped with optimized detectors for their specific geo-fenced areas. While it may be infeasible for these local "experts" to directly share their raw sensor data or detectors (*e.g.*, due to data size and format; commercial and intellectual properties; implementation incompatibility), the *predictions* (*e.g.*, detected 3D bounding boxes) are more lightweight and standardized. Several recent works also show that sharing predictions would benefit each participating agent's perception accuracy (Xu et al., 2023; 2022b; Yu et al., 2022; Chen et al., 2019; Xu et al., 2022a; Wang et al.,

---

[1]Project page: **https://jinsuyoo.info/rnb-pop**.

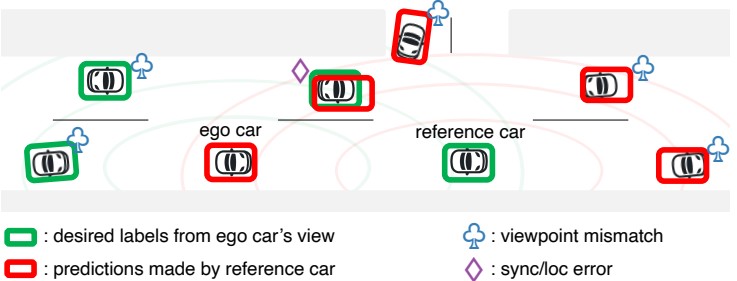

Figure 1: **Research problem of *learning from others' predictions*.** We study the scenario where an agent (*e.g.*, ego car) leverages the predictions made by another agent (*e.g.*, a high-end reference car) as supervision to train its own 3D object detector. We observe two challenges: (1) viewpoint mismatch between two cars and (2) mislocalization due to synchronization/GPS errors.

2020a; Lu et al., 2024; Hong et al., 2024; Hu et al., 2024), further incentifying such a collaborative scenario. Last but not least, sharing predictions implies that there is no need for all the agents to use the same sensors. An agent adopting a new sensor or entering an unfamiliar environment thus could borrow the predictions made by other agents, potentially equipped with higher-end sensors, as labels to train its detector. (Please see Sec. 3.1 for a feasibility and practicality discussion of our setting.)

In this paper, we thus investigate a new scenario to construct a 3D object detector: *learning from the predictions of a nearby agent equipped with an accurate detector.* We use the real-world collaborative driving dataset (Xu et al., 2023) as the testbed. For each 3D road scene, this dataset records two LiDAR point clouds from two nearby cars distancing between $0 \sim 100$ meters and offers object labels separately for each point cloud. We use one of the cars as the reference agent, equipped with an accurate 3D detector, to provide predicted labels from which the other (ego) car can learn.

At first glance, this research problem may appear trivially as a standard supervised learning problem — using another agent's predictions as labels to train the detector for the ego car. However, our preliminary attempt showed that this straightforward approach results in poor performance. We identify two major challenges (Fig. 1). First, in real-world applications, inaccuracies such as GPS errors and synchronization delays between agents are common. For example, a minor delay of just 0.1 seconds can cause a discrepancy of several meters in localization for a vehicle traveling at 60 mph. Second, the viewpoints of the two agents can vary significantly. An object visible to one agent might be obscured or out of range for the other due to occlusion or distance, leading to false positives and negatives in the predictions. Training with such *mislocalized* and *viewpoint-mismatched* labels inevitably results in suboptimal performance for the new 3D detector of the ego car.

To address these challenges, we propose a learning pipeline termed as ***R**efining **&** Discovering **B**oxes for 3D **P**erception from **O**thers' **P**redictions* (R&B-POP). For mislocalization, we train a box refinement module to rank the noisy candidates and correct their locations. Notably, this module requires very few human labels (1% or less), or even no human labels if simulation data are available. We also develop a coarse-to-fine approach to search for high-quality candidates around the predicted object locations efficiently, tackling large localization errors. For viewpoint mismatch that results in false negatives in the ego car's perspective, we present an effective self-training strategy empowered by a novel distance-based curriculum, enabling the detector to first learn from a subset of high-quality labels and in turn fill in the missing labels for the model to continually learn from. With these approaches, we significantly improve the quality of pseudo labels and, consequently, produce a much more accurate 3D detector for the ego car, with very limited human labeling — the Average Precision (AP) at IoU 0.5 increases from 22% to 56.5% using only 40 labeled frames!

In summary, we introduce a novel research problem that learns 3D perception for a new agent with reference agent's predictions. We identify the main challenges about the label quality and propose corresponding solutions. With extensive experiments, we demonstrate the applicability of the new learning scenario as well as the improvements achieved by our designs.

## 2 RELATED WORK

**3D object detection** serves an important role in real-world applications such as autonomous driving. The detector takes 3D signals (*e.g.*, LiDAR points) as input, and predicts the existence and the

location of objects of interest. Notable development has been made thanks to the recently curated large datasets (Geiger et al., 2012; Caesar et al., 2020; Sun et al., 2020; Yu et al., 2022; Xu et al., 2023). The existing approaches can be categorized as voxel-based (or pillar-based) methods (Zhou & Tuzel, 2018; Yan et al., 2018; Lang et al., 2019), which subdivide irregular 3D point space into regular space, and point-based methods (Zhao et al., 2021; Yang et al., 2018), which directly extract discriminative point-wise features from the given point clouds. Regardless of approaches, these methods require manually annotated data of high-quality to achieve satisfactory performance. In this study, we aim to bypass such a labeling cost and demonstrate our new label-efficient learning method with representative 3D detectors (Lang et al., 2019; Yan et al., 2018).

**Label-efficient learning.** Self-supervised learning is a promising way to bypass extensive label annotation (Chen & He, 2021; He et al., 2020; Chen et al., 2020; Hjelm et al., 2018). Pre-trained with abundant, easily collectible unlabeled data, the detector backbone is shown to largely reduce the labeled data for fine-tuning (Pan et al., 2024; Yin et al., 2022; Xie et al., 2020b). Label-free 3D object detection from point clouds has gained attention due to its effective data utilization (You et al., 2022a;b; Luo et al., 2023; Najibi et al., 2022; Zhang et al., 2023b; Choy et al., 2019; Baur et al., 2024; Seidenschwarz et al., 2024; Yang et al., 2021) and generalization beyond specific class information during training (Najibi et al., 2023). Researchers have also explored semi-supervised methods (Wang et al., 2021; Zhang et al., 2023a; Liu et al., 2022a; Xia et al., 2023; Yang et al., 2024; Xia et al., 2024; Liu et al., 2022b) or offboard detectors (Qi et al., 2021; Ma et al., 2023) to reduce the manual labeling efforts. Orthogonally, we study a new scenario to learn the detector in a label-efficient way by considering *beyond a single source of information*. Specifically, the predictions from well-trained detectors of reference units near the ego car are leveraged as (pseudo) labels.

**Domain adaptation.** Our setting is related to domain adaptation (DA), as we aim to improve an object detector in a new environment (*e.g.*, a new location or data pattern). Existing studies (Wang et al., 2020b; Chen et al., 2024; Yang et al., 2021; 2022) mostly focus on the generic, single-agent unsupervised DA setting, while a few leverage application-specific cues, *e.g.*, repetitions (You et al., 2022c), to facilitate adaptation. Our setting belongs to the second branch, in which we explore a multi-agent scenario. Our goal is not to compete with the generic setting. Instead, generic DA techniques, *e.g.*, advanced self-training (Peng et al., 2023; Wang et al., 2023; Hegde et al., 2023; Tsai et al., 2023), can be compatible with our setting to further boost the performance.

**Curriculum learning.** Many studies have shown that properly ordering the data to progressively add harder samples during training leads to superior performance. The so-called "curriculum learning" (Bengio et al., 2009) has also been explored in object detection (Li et al., 2017; Sangineto et al., 2018). For LiDAR-based 3D detection, researchers have applied the concept for better data augmentation during training (Yang et al., 2021; Zhu et al., 2023). We investigate the task-specific characteristics from the data and discover a meaningful correlation between the label quality and the ego-car-reference-car distance. We then apply this observation to design an effective training curriculum.

**Collaborative perception.** To mitigate limited detection range and occlusion, self-driving researchers have recently focused on integrating nearby detectors' information (Chen et al., 2019; Xu et al., 2022b;a; 2023; Yu et al., 2022; Wang et al., 2020a; Lu et al., 2024; Hong et al., 2024; Hu et al., 2024). During inference, more than one detector communicates with each other and shares their information (*e.g.*, input signal, feature, or predicted boxes) to detect objects better. While also leveraging other cars' information, our research focus is different — we investigate a new label-efficient learning scenario, using other (expert) cars' predictions as supervision to build the ego car's detector offline.

## 3 LEARNING 3D PERCEPTION FROM OTHERS' PREDICTIONS

We study a novel research problem in autonomous driving: training a 3D detector using bounding boxes supplied by a nearby agent. This scenario, while unexplored, can reduce or even eliminate labeling efforts. We identify the key challenges and propose the learning pipeline to address them.

### 3.1 PROBLEM DEFINITION AND FEASIBILITY

**Problem setup.** Without loss of generality, we assume that around the ego car (*i.e.*, $E$), there is a reference car (*i.e.*, $R$) equipped with an accurate 3D object detector $f_R$. $E$ and $R$ are both equipped with 3D sensors (*e.g.*, LiDAR) and collect their point clouds (*i.e.*, $X_E$ and $X_R$) in the same road scene. Notice that $X_E$ and $X_R$ can have different patterns due to variations in hardware. $R$, the car that $E$ learns from, share 3D bounding boxes of foreground objects in the global coordinate from its

Table 1: **Label quality** in recall and precision at IoU 0.5 with $E$'s GT. Our methods improve the label quality significantly.

| | pseudo label | $R$'s GT rec. / prec. | $R$'s pred rec. / prec. |
|---|---|---|---|
| ① | initial boxes | 54.8 / 43.2 | 56.1 / 48.0 |
| ② | + basic filtering | 54.1 / 65.6 | 55.3 / 71.4 |
| ③ | **+ our refinement** | **66.2 / 85.4** | 65.2 / 79.0 |
| ④ | **+ our self-train** | **72.5 / 90.0** | 74.4 / 87.7 |
| ⑤ | sharing detector | - | 78.8 / 86.8 |

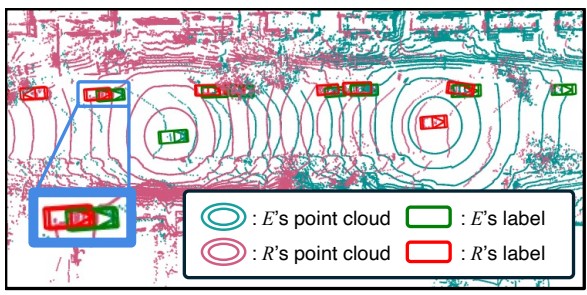

Figure 2: **Point and box discrepancies** between ego and reference cars on the real dataset (Xu et al., 2023).

detector, *i.e.*, $Y_R = f_R(X_R)$. Our goal is to train a 3D detector $f_E$ that works with $X_E$, by using $Y_R$. Please see Sec. S1 for more detailed problem setup.

**Feasibility and practicality.** Before proceeding, we consider two critical questions, "Why can nearby agents obtain accurate detectors?" and "Why can they not directly share their detectors?" Besides the examples mentioned in Sec. 1 (*e.g.*, self-driving taxis), we emphasize that these nearby agents need not be "omniscient." Instead, they only need to be experts in geo-fenced areas where the ego agent passes by and can even be static, making training their detectors easier and much more label-efficient, *e.g.*, using the repetition or background cues (You et al., 2022b;a; Dao et al., 2024).

Regarding "why these agents cannot just share their detectors," we note that while open-sourcing is common in the research community, there are many considerations and constraints when it comes to practical scenarios. First, the ego and the reference agents do not need to have the same sensors. Indeed, they may not even perceive the environment from the same views, *e.g.*, the reference agent can be a roadside unit placed six meters high and facing down (Yang et al., 2023; Dao et al., 2024). This discrepancy makes the direct deployment of the reference agent's model to the ego agent suboptimal. Second, the two agents may be equipped with different computational platforms, *e.g.*, the reference one is equipped with GPUs while the ego one with FPGA boards and hardware acceleration code (Hao et al., 2019; Hao & Chen, 2021), making direct deployment more challenging. Last but not least, reference agents' detectors may be specifically designed and trained, *e.g.*, using private data. Sharing them thus raises intelligent property or privacy concerns. Putting things together, we argue that our setting is realistic and has significant practical implications.

### 3.2 CHALLENGE

**First attempt.** We use the recently released real-world collaborative driving dataset (Xu et al., 2023) as the testbed. For each 3D road scene (with a time tag), the dataset provides LiDAR point clouds and ground-truth 3D bounding boxes from each agent's perspective. (We keep the data and experimental details in Sec. 4.) We begin by employing $Y_R$ (*i.e.*, $R$'s predictions) directly as labels for $E$ to train $f_E$, after transforming $Y_R$ into $E$'s coordinate system. To establish an upper bound, we also train a detector using $E$'s ground-truth labels. The result shows that the detector performance by naively using $Y_R$ is way much worse than the upper bound (AP at IoU 0.5: ① 22.0 *vs.* ♣ 58.4 in Table 2).

At first glance, such a gap, with no doubt, must come from reference car $R$'s prediction errors. To eliminate the effect, we use $R$'s ground-truths as labels (*i.e.*, $R$'s GT) to train another detector for the ego car $E$. To our surprise, using $R$'s GT can hardly improve the detector's performance, suggesting the existence of other, more fundamental factors in the real-world environment.

**Key challenges.** To search for the root cause of the poor detector performance, we visualize the point clouds and ground-truth bounding boxes of the two cars in Fig. 2. We identify two major sources of errors: viewpoint mismatch and mislocalization. Viewpoint mismatch occurs when objects are obscured from *one* sensor's view due to occlusion or field of view limitations, while mislocalization results from GPS inaccuracies and synchronization delays. For instance, a communication delay of 0.1 seconds in a car traveling at 60 mph can result in a localization discrepancy of 2.7 meters. These errors can significantly degrade the quality of the learned detector $f_E$ for the ego car $E$ — the training labels are simply *noisy*. To further dive into these challenges, we measure the precision and recall of $Y_R$ *vs.* the ego car $E$'s ground-truth labels to assess label quality, as shown in Table 1. Even after applying basic filtering commonly used in autonomous driving (*e.g.*, removing distant boxes or those

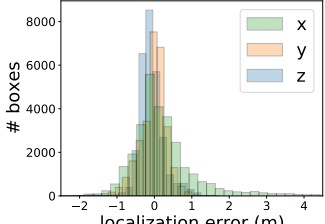

Figure 4: **Box ranker for refining localization error.** With a few annotated frames (or boxes), we train a ranker that can estimate the quality of a given box. During inference for pseudo labels, we sample multiple candidates near the initial noisy box and choose the one with the best IoU.

with few points that are beyond $E$'s field of view), the label quality remains unsatisfactory (Table 1 ②). In the following sections, we introduce our pipeline R&B-POP to tackle these challenges.

### 3.3 LABEL-EFFICIENT BOX REFINEMENT

**Preliminaries.** We conduct a detailed analysis of the localization discrepancies in each coordinate (x forward, y leftward, z upward) between $R$'s and $E$'s overlapping ground-truth boxes, as illustrated in Fig. 3. Notice that a mere 0.5-meter discrepancy in the x and y coordinates can drastically reduce the IoU from 100% to 30%. Training with such inaccurate pseudo labels inevitably leads to suboptimal performance in $E$'s 3D detector. A refinement module for the labels is thus necessary!

**Baseline approach with heuristics.** To begin with, we adopt the algorithm proposed in Luo et al. (2023), which refines boxes using heuristics. Specifically, multiple boxes are sampled around the initial noisy boxes, and the optimal boxes are selected based on the best alignment of edges and sizes between the boxes and point clouds. However, this method requires certain conditions to achieve satisfactory performance, such as multiple trajectories at the same location, potentially limiting the applicability. As in Table 2, adapting it to our problem brings marginal gains, especially for the high IoU of 0.7 (AP ① 4.2 *vs.* ③ 10.3).

Figure 3: **Mislocalization** between $E$'s and $R$'s GT.

**Label-efficient box ranker.** To address this limitation, we propose to train a *box ranker* that evaluates the localization quality of given bounding boxes. Instead of predicting a 3D box from scratch (*i.e.*, a typical detection problem), learning to select and adjust among noisy candidates is a much easier task. *We thus expect learning such a ranker needs much fewer labeled data!* To investigate this idea, we sample a handful of $E$'s point clouds with ground-truths to train the ranker. We randomly sample multiple boxes around each annotated object box and crop point clouds outlined by those sampled boxes (with expansion). The training objective is dual: to regress the IoU between a sampled box and the annotated box, serving as the indicator of localization quality, and to estimate the offset to the annotated box, further refining its location. During inference, we use $Y_R$ as initial boxes and sample $N$ boxes around each. The top-ranked boxes are selected among all candidates to construct $Y'_R$ as pseudo labels for training the 3D detector $f_E$ for $E$ (see Fig. 4). We adopt a neural network similar to PointNet (Qi et al., 2017) for the ranker for its simplicity. Please see the supplementary for details.

**Coarse-to-fine (C2F) refinement.** As previously discussed, minor time delays can result in large discrepancies of several meters. To address this and expand the search region, thereby increasing the number of high-quality bounding box candidates for our ranker, we employ a two-stage approach during *inference*, as illustrated in Fig. 5. In the first stage, we generate $\frac{N}{2}$ candidate boxes for each initial box by sampling translations from a wider range using a uniform

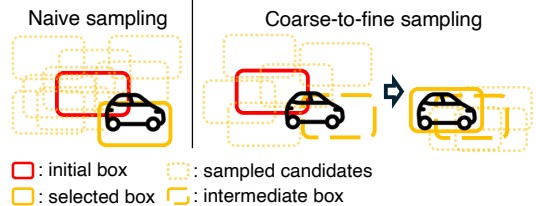

Figure 5: **Sampling methods for box refinement.** Proposed C2F is more effective in large mislocalization.

distribution, while keeping the scale and pose of the boxes fixed. In the second stage, we select the top-$K$ boxes from the first stage to serve as new initials and sample another $\frac{N}{2}$ total new boxes around them, this time considering all degrees of freedom (*i.e.*, translation, scale, and pose) but from a narrower range using normal distributions. This coarse-to-fine (C2F) strategy effectively bridges the large localization gap and enhances the refinement quality of $Y'_R$. With the box ranker and C2F, we raise the label quality from a recall of 55.3 to 65.2 and a precision of 71.4 to 79.0 upon basic filtering,

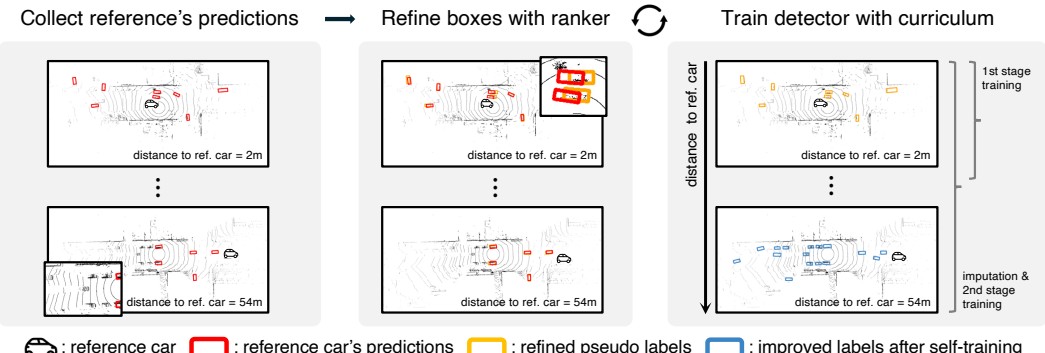

: reference car    : reference car's predictions    : refined pseudo labels    : improved labels after self-training

Figure 6: **Overall pipeline of R&B-POP.** The ego car first receives reference's predictions which contain inherent noises (Sec. 3). It refines their localization with proposed box ranker (Sec. 3.3). Then, it creates high-quality pseudo labels by distance-based curriculum for self-training (Sec. 3.4).

as shown in Table 1 ② *vs.*③, *using only 40 labeled frames*. Consequently, the performance of $f_E$ also shows a significant gain from 22.0% to 38.0% in AP at IoU 0.5, as reported in Table 2 ① *vs.*④.

**Ranker-based filtering.** The trained ranker not only refines the given boxes but also estimates their IoU with ground-truth boxes. Applying a threshold on predicted IoU effectively removes false positives, thus improving detection performance as shown in Table 4b.

## 3.4 DISTANCE-BASED CURRICULUM

Viewpoint mismatch introduces false positives (*i.e.*, objects should not be visible from *E*'s perspective) and negatives (*i.e.*, objects should be visible to *E* but are not provided by *R*) in $Y_R$. While false positives can be removed by filtering (*e.g.*, basic and ranker-based filtering), false negatives are much harder to be recovered. It becomes necessary to discover new boxes from *E*'s perspective.

**Box discovery from the ego car *E*'s view.** Inspired by the previous work (You et al., 2022b), self-training (Lee et al., 2013; Xie et al., 2020a) is a popular technique to propagate labels to unlabeled data. This method typically employs high-quality labels to train the base detector and subsequently uses its predictions to generate new pseudo labels for further fine-tuning cycles. However, as discussed in previous sections, our initial (pseudo) labels are inherently noisy, which can hinder the efficacy of self-training. This leads to a pivotal question: *How to ensure the quality of pseudo labels for effective self-training?*

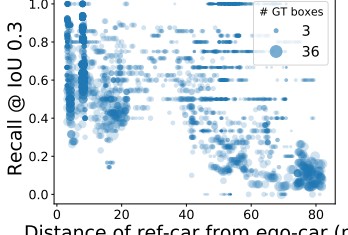

Figure 7: **Quality of pseudo labels from *R*'s predictions** drops when two cars are farther apart.

**Key observation about distance.** We find out that there exists a unique property in our learning scenario — the extent of viewpoint mismatch is correlated with the distance between *E* and *R*. Specifically, discrepancies are typically reduced when the two are closer and increased when they are distant (Fig. 7). This intuition leads us to use the distance of two cars as an indicator of the quality of pseudo labels provided by *R*. Building on this observation, we develop two distance-based methods in the following.

**Distance-based curriculum for self-training.** We create a high-quality subset of pseudo labels by applying a simple distance threshold $T_{E\text{-}R}$ to all frames, meaning that we trust pseudo labels from *R* when two cars are close enough. In the first round of self-training, the 3D detector $f_E$ is exclusively trained on this high-quality subset. In later rounds, we fine-tune on *all frames* with pseudo labels predicted by $F_E$. This approach propagates labels learned from confident frames to unconfident ones.

**Distance-based filtering.** Self-training needs a filtering mechanism to select high-quality predictions by the current detector, which are then treated as true labels to supervise the next round of detector training. Normally, this is done by setting a fixed threshold $T_c$ in prediction confidence[2]. Here, we employ a distance-based threshold, inspired by our self-training procedure. Specifically, since we trust frames with smaller ego-car-to-reference-car distances and train the detector with them in the first round, the detector will inherently be overly confident in these frames. As such, a higher threshold

---

[2]We note that the detector's confidence is not the same as the IoU predicted by the ranker in Sec. 3.3.

shall be assigned when selecting pseudo labels from them. We implement this idea by increasing the confidence threshold with a negative linear function of the distance (*i.e.*, $T_c + \lambda/distance(E, R)$).

Put together, these two distance-based approaches not only uncover boxes that should be visible to $E$ but also preserve the quality of pseudo labels for self-training. As shown in Table 1 ④, self-training with distance-based curriculum further improves label quality from 79.0% to 87.7% in precision and 65.2% to 74.4% in recall, resulting in an enhancement of $f_E$'s performance from AP of 38.0% to 56.5%, as detailed in Table 2 ④ *vs.*⑩. As a reference, we show the predicted label quality on $X_E$ using $f_R$ in Table 1 ⑤, simulating the ideal case where the object detector can be shared. Our R&B-POP achieves similar label quality, demonstrating the applicability of our setting for learning high-quality detectors from others' predictions.

## 3.5 OVERALL PIPELINE

Putting everything together, our overall *offline* pipeline involves the following steps (Fig. 6).

**Step 0. Ranker training** with few annotated labels (Sec. 3.3).

**Step 1. First-round self-training:** Preparing pseudo labels after receiving $R$'s predictions (applying basic filtering), further improving labels with ranker + C2F and ranker-thresholding (Sec. 3.3), and then training the detector $f_E$ with closer frames (Sec. 3.4).

**Step 2. Second-round self-training:** Preparing pseudo labels after receiving $f_E$'s prediction (applying distance-filtering in Sec. 3.4), further improving labels with ranker + C2F and ranker-thresholding (Sec. 3.3), and then training the detector $f_E$ with all frames.

## 4 EXPERIMENTS

### 4.1 EXPERIMENTAL SETUPS

**Datasets.** To validate the effectiveness of our method, we conduct experiments primarily on the V2V4Real dataset (Xu et al., 2023), which consists of 40 clips with a total of 18k frames by driving two cars, Tesla and Honda, together within 100m. LiDAR points are acquired with a Velodyne VLP-32 LiDAR sensor. The dataset provides annotations for different types of vehicles, such as cars and trucks. (Please see additional results on the OPV2V dataset (Xu et al., 2022c) in the supplementary.)

To align with our research purpose, we re-split the original data into three portions: "$R$ pretraining", "$R$ prediction/$E$ training", and "$E$ validation/test" (Fig. 8). Specifically, we split them into two subsets containing 20 clips and use the first subset to pre-train $R$'s detector $f_R$. Then we inference on the second subset to provide pseudo labels $Y_R$ for training $E$'s detector $f_E$ together with $E$'s point

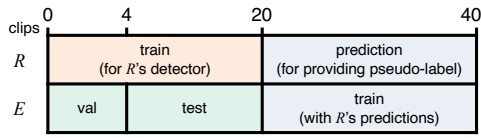

Figure 8: **Dataset split.** We re-split V2V4Real (Xu et al., 2023) for our setting.

clouds. We validate and test the $E$'s performance on the first subset by splitting it into 20% and 80%. Our re-split gives 4,488 frames for $R$ pretraining, 4,463×2 frames for $E$ training, and 870 and 3,618 frames for $E$ validation/test respectively. The performance in the paper is reported on $E$'s test set.

**Evaluation.** We follow Xu et al. (2023) to merge different types of vehicles (*e.g.*, cars, trucks) into a single category [3]. We report the average precision (AP) of detectors in the bird's-eye view with IoU thresholds of 0.5 and 0.7. Specifically, we set the region of interest to [-80, 80]m for the heading direction and [-40, 40]m for the direction perpendicular to the moving direction. We also report the AP on different depth ranges [0-30, 30-50, 50-80, 0-80]m following Luo et al. (2023).

**Implementation.** We conduct experiments with PointPillars (Lang et al., 2019) as a default **detector**. We train it with 60 epochs and a batch size of 64 on 8 NVIDIA Tesla P100 GPUs. We use Adam optimizer and an initial learning rate of 2e-3 dropped to 2e-5 by cosine annealing decaying strategy (Loshchilov & Hutter, 2017). For training the **box ranker**, a PointNet (Qi et al., 2017) specified

---

[3]We note that V2V4Real (Xu et al., 2023) does not label objects beyond vehicles and the data distributions across different types of vehicles are largely imbalanced. Thus, it is infeasible to study multi-class vehicle detection. That said, extending our approach to a multi-class setup would be straightforward if a suitable dataset is available. The key is to make the ranker category aware. Please refer to our experiments on extending the ranker to a multi-class setting in the supplementary.

Table 2: **Main results: validation of the proposed learning scenario and methods.** The results indicate a new research problem of *learning with others' predictions* has inherent challenges. With proposed R&B-POP, we significantly close the gap to the upper bound that directly uses ego car's ground-truth labels (⑩ 56.5 *vs.*♣ 58.4). The performance is reported on PointPillars (Lang et al., 2019) with 32-beam LiDAR.     : uses GT labels.     : our proposed methods.

| | pseudo label | box refinement | self-training | AP @ IoU 0.5 | | | | AP @ IoU 0.7 | | | |
|---|---|---|---|---|---|---|---|---|---|---|---|
| | | | | 0-30m | 30-50m | 50-80m | 0-80m | 0-30m | 30-50m | 50-80m | 0-80m |
| ① | *R*'s pred | - | - | 34.7 | 13.5 | 8.6 | 22.0 | 7.6 | 2.2 | 1.5 | 4.2 |
| ② | *R*'s GT | - | - | 29.7 | 14.1 | 7.3 | 19.6 | 5.9 | 2.2 | 1.8 | 3.7 |
| ③ | *R*'s pred | heuristic (Luo et al., 2023) | - | 53.2 | 22.0 | 16.9 | 37.8 | 16.5 | 4.5 | 3.9 | 10.3 |
| ④ | *R*'s pred | ranker | - | 50.3 | 24.7 | 18.2 | 38.0 | 33.6 | 12.9 | 8.9 | 22.9 |
| ⑤ | *R*'s pred | - | naive (You et al., 2022b) | 45.9 | 18.7 | 16.5 | 32.4 | 13.8 | 4.6 | 5.9 | 9.2 |
| ⑥ | *R*'s pred | heuristic (Luo et al., 2023) | naive (You et al., 2022b) | 50.4 | 19.6 | 15.4 | 35.4 | 13.8 | 4.2 | 3.6 | 8.9 |
| ⑦ | *R*'s pred | ranker | naive (You et al., 2022b) | 60.6 | 29.7 | 19.2 | 45.0 | 40.8 | 16.7 | 9.5 | 28.0 |
| ⑧ | *R*'s pred | - | distance-based curriculum | 57.3 | 29.6 | 21.0 | 42.5 | 21.0 | 5.9 | 3.9 | 12.7 |
| ⑨ | *R*'s pred | heuristic (Luo et al., 2023) | distance-based curriculum | 60.5 | 25.5 | 17.0 | 43.2 | 18.3 | 4.4 | 3.0 | 11.0 |
| ⑩ | *R*'s pred | ranker | distance-based curriculum | 73.3 | 43.3 | 23.3 | 56.5 | 47.1 | 21.1 | 10.0 | 32.6 |
| ♣ | *E*'s GT | - | - | 75.2 | 45.9 | 28.8 | 58.4 | 51.7 | 25.4 | 14.8 | 36.3 |

🚗 : reference car

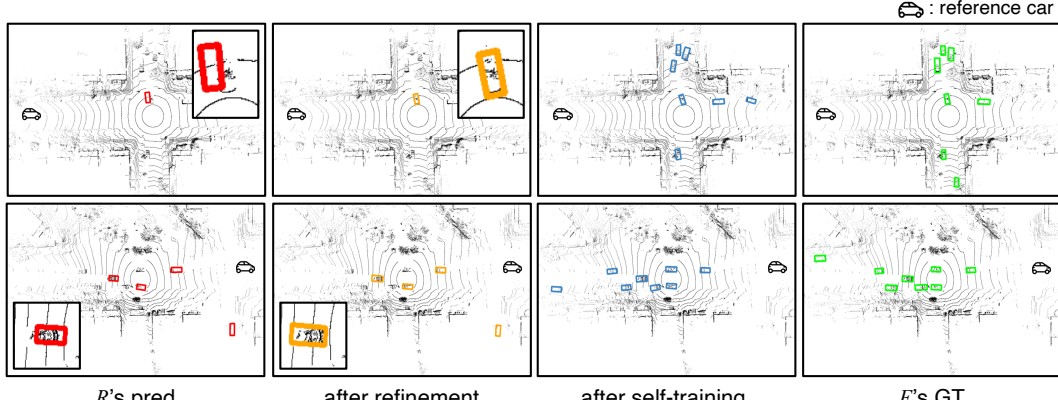

*R*'s pred      after refinement      after self-training      *E*'s GT

Figure 9: **Qualitative results.** The quality of pseudo labels is gradually improved with the proposed R&B-POP. Our ranker successfully fixes mislocalization errors, and distance-based curriculum further discovers new objects from *E*'s view.

in the supplementary, we use 40 annotated frames ($< 1\%$ of training data) to generate 11k samples. In the *offline* ranker inference, we sample $N = 512$ boxes around each prediction provided by the reference car. We first sample 256 boxes in the coarse stage, select top-3 boxes, and then sample the remaining 256 boxes near the selected boxes in the fine stage. For curriculum learning, we set $T_{E\text{-}R}$ to 40m. Also, we set the ranker threshold to 0.5, and $\lambda$ for the distance-based threshold to 1 with a fixed confidence threshold $T_c$ of 0.2. Please refer to the supplementary material for more details.

## 4.2 EXPERIMANTAL RESULTS

We first demonstrate our scenario, *learning with others' predictions*, with a basic setup: *R* and *E* both have 32-beam sensors with PointPillars (Lang et al., 2019), but only *R*'s detector was pre-trained. Table 2 compares different methods for pseudo labels, including baselines such as standard self-training (You et al., 2022b) and heuristic-based label refinement (Luo et al., 2023). For fair comparisons, we note that we only utilize annotated boxes from 40 frames to train our ranker, not to train the detector. For the heuristic-based refinement (Luo et al., 2023), as our dataset has no repeated traversal to estimate movable objects, we use RANSAC (Fischler & Bolles, 1981) instead.

Firstly, our ranker brings a notable gain over heuristics-based refinement (③ 10.3 *vs.* ④ 22.9, ⑥ 8.9 *vs.* ⑦ 28.0, and ⑨ 11.0 *vs.* ⑩ 32.6 on AP at IoU 0.7), demonstrating its effectiveness to address mislocalization. Secondly, our distance-based curriculum consistently improves the performance over the standard self-training (⑤-⑦ 32.4/35.4/45.0 *vs.*⑧-⑩ 42.5/43.2/56.5), demonstrating the necessity of using higher-quality samples for self-training. Finally, by comparing with detectors trained with *E*'s ground-truth, our method achieves on par with the upper bound (⑩ 56.5/32.6 *vs.*♣ 58.4/36.3).

Table 3: **Ablations on box ranker.** (a) The training of the ranker requires very few human labels. Using simulation data further eliminates the need yet performs on par. (b) The proposed inference strategies effectively generate high-quality pseudo labels, resulting in better performance.

| training data | # annot. frames | AP @ IoU 0.5 | AP @ IoU 0.7 |
|---|---|---|---|
| | 20 | 54.8 | 30.0 |
| $E$'s GT | 40 | 56.5 | 32.6 |
| | 80 | 55.5 | 32.8 |
| Simulation (Xu et al., 2022c) | 21k | 52.2 | 28.4 |

(a) Analysis on training labels.

| | offset | naive | C2F | AP @ IoU 0.5 | AP @ IoU 0.7 |
|---|---|---|---|---|---|
| ① | ✓ | | | 50.2 | 28.5 |
| ② | ✓ | ✓ | | 56.5 | 30.7 |
| ③ | ✓ | | ✓ | 56.5 | 32.6 |
| ④ | | | ✓ | 54.4 | 31.0 |

(b) Analysis on inference strategies.

Table 4: **Ablations on curriculum self-training.** The results show that each individual component in our proposed method contributes to the optimal performance. (a) Our curriculum selectively leverages useful frames during each stage of training. (b) Our label thresholding effectively discards noisy labels for training, resulting in improved performance.

| | stage 1 | stage 2 | AP @ IoU 0.5 | AP @ IoU 0.7 |
|---|---|---|---|---|
| ① | 0-90m | 0-90m | 45.0 | 28.0 |
| ② | 0-40m | 0-40m | 50.3 | 24.7 |
| ③ | 0-40m | 40-90m | 51.1 | 26.0 |
| ④ | 40-90m | 40-90m | 33.5 | 20.0 |
| ⑤ | 0-40m | 0-90m | 56.5 | 32.6 |

(a) Analysis on different curriculums.

| ranker threshold | distance-based threshold | AP @ IoU 0.5 | AP @ IoU 0.7 |
|---|---|---|---|
| | | 50.2 | 26.2 |
| | ✓ | 54.9 | 30.3 |
| ✓ | | 52.2 | 28.3 |
| ✓ | ✓ | 56.5 | 32.6 |

(b) Analysis on box filtering.

Fig. 9 visualizes the improvement of pseudo labels with our method. The ranker successfully refines mislocalized pseudo labels provided by $R$. Moreover, our distance-based curriculum discovers new bounding boxes that $E$ couldn't receive from $R$'s perception, without introducing many false positives.

### 4.2.1 ABLATION STUDY

**Analysis on ranker training.** We first check the performance of the ranker trained with different number of annotated frames (*i.e.*, 20, 40, and 80) in Table 3a. Notably, we observe a performance boost when the ranker is trained with 40 frames compared to 20 frames, and the performance gain diminishes with more than 40 frames. This demonstrates that the ranker already perform well with very few labeled data (*i.e.*, 1% of total). Moreover, we train our ranker with 21k samples generated by CARLA simulator (Xu et al., 2022c) and achieves on par performance, exploring to remove the need of human labels. As shown, our ranker trained with off-the-shelf simulated data can achieve 28.4% AP at IoU 0.7, higher than the 11.0% AP at IoU 0.7 achieved by Luo et al. (2023) (Table 2 ⑨).

**Analysis on ranker inference.** We investigate the impact of inference strategies for the refinement on the final detection performance in Table 3b. The results indicate that sampling boxes (① 28.5 *vs.* ② 30.7) and coarse-to-fine refinement (② 30.7 *vs.* ③ 32.6) contribute to superior performance, especially for the fine-grained quality of IoU 0.7. We also observe the benefit of using predicted offsets to further refine box locations (④ 31.0 *vs.* ③ 32.6). This demonstrates the effectiveness of our sophisticated box refinement strategies.

**Analysis on curriculum for self-training.** In our method, we split the training data into two subsets with the threshold $T_{E\text{-}R}$ of 40m. We train the detector on different combinations of subsets and check its performance (Table 4a). We observe that using initial low-quality frames to train the model gives significantly lower performance (④ 33.5 *vs.* ⑤ 56.5). Also, we see that the performance escalates as we utilize more high-quality frames during the next self-training stage (② 50.3, ③ 51.1 *vs.* ⑤ 56.5), verifying the effectiveness of our curriculum choice.

**Analysis on box filtering for self-training.** To prevent the model from introducing false positive boxes during the self-training, we design a distance-based confidence threshold and ranker-based filtering. As shown in Table 4b, the detector improves with our strategies. This indicates a good balance between recall and precision provided by our method, resulting in better detection performance.

Table 5: **Application to different scenarios.** We study cases where $E$ and $R$ have different LiDAR patterns or detector architectures. We further equip $E$ with a pre-trained detector aiming to adapt to $R$'s driving scenes. The results indicate the flexibility of the new research problem of *learning from others' predictions*. [†]We use simulation data collected by Xu et al. (2022c).

**(a) Different sensors.**

| # beams | | AP @ IoU | |
| --- | --- | --- | --- |
| $R$ | $E$ | 0.5 | 0.7 |
| 8 | 32 | 54.8 | 31.6 |
| 16 | 32 | 56.0 | 31.5 |
| 32 | 32 | 56.5 | 32.6 |

**(b) Different detectors.**

| $E$'s detector | $R$'s detector | AP @ IoU | |
| --- | --- | --- | --- |
| | | 0.5 | 0.7 |
| SECOND | SECOND | 56.2 | 37.2 |
| | PointPillars | 56.7 | 37.3 |
| | $E$'s GT | 63.8 | 43.9 |
| PointPillars | PointPillars | 56.5 | 32.6 |
| | SECOND | 54.4 | 32.4 |
| | $E$'s GT | 58.4 | 36.3 |

**(c) Domain adaptation.**

| | | AP @ IoU | |
| --- | --- | --- | --- |
| | label from | 0.5 | 0.7 |
| pre-trained (PT) | CALRA[†] | 51.7 | 32.5 |
| fine-tuned | PT | 58.0 | 35.0 |
| | PT + $R$'s pred | 60.8 | 37.7 |

### 4.2.2 APPLICATIONS TO DIFFERENT SCENARIOS

R&B-POP can also be used when $E$ and $R$ have different detectors and hardware sensors, and are across different domains. This section evidences the flexibility of our method.

**Different sensors.** To show that our pipeline can be trained on facilities with different sensors, we conduct experiments with synthesized 16- and 8-beam LiDAR point clouds with the beam-dropping algorithm (Wei et al., 2022). Specifically, we assume $R$ has either 8-, 16-, and 32-beam, respectively, and the detector architecture is fixed to PointPillars (Lang et al., 2019). As shown in Table 5a, the detection performance does not drop when using less advanced sensors (*i.e.*, fewer beams). This indicates the flexibility of our algorithm in different sensor configurations.

**Different detectors.** We conduct experiments to demonstrate the effectiveness of our pipeline on different detectors, *e.g.*, PointPillars (Lang et al., 2019) and SECOND (Yan et al., 2018), by using various combinations in Table 5b. The results show that the detector performance of $E$ does not rely on $R$'s detector, which implies $E$ tends to learn agnostically to $R$'s setups. Together with the experiments for different sensors, this highlights the general flexibility of our method.

**Different domains.** We explore our algorithm's applicability to the domain adaptation scenario with a synthetic to real case. In doing so, we assume that $E$'s detector has been pre-trained on simulation data (Xu et al., 2022c). As shown in Table 5c, the performance of $E$'s pre-trained detector improves with adaptation to real domains (via self-training). Notably, we compare adaptation with and without using $R$'s predictions and observe that leveraging the predicted boxes from $R$ is beneficial.

### 4.2.3 ADDITIONAL EMPIRICAL STUDIES

We leave additional results in the supplementary, including the ideal scenario where the object detector can be shared, analysis on ranker and self-training, and extension to another dataset.

## 5 CONCLUSION AND DISCUSSION

In this work, we have introduced *learning with others' predictions*, a new way to train a 3D detector with the predictions of reference units. We have systematically identified the inevitable task-specific problems: false positive, false negative, and noisy boxes due to either viewpoint mismatch or synchronization/GPS errors. Next, we have proposed to improve the quality of pseudo labels by two solutions: A box ranker and distance-based curriculum self-training. We have demonstrated a wide applicability of our learning scenario with different detectors, sensors, and domains.

**Limitations and future work.** The ego car's detector can benefit from other reference units, such as roadside units in smart cities. In our future work, we plan to investigate more diverse scenarios. Moreover, the ego car and the reference car using different modalities would be a direction to explore further. We believe that our findings and approach have set the foundation for it. At a high level, our approach is sensor and modality-agnostic. Regardless of the type of sensors and detectors (camera-based or LiDAR-based) used, if we aim at 3D perception, they will produce 3D bounding boxes as pseudo labels. Our method does not necessitate a specific model for providing the pseudo labels and can be easily adapted to various sensor types. We leave this extension to future work.

ACKNOWLEDGMENT

This research is supported in part by grants from the National Science Foundation (IIS-2107077, IIS-2107161). We are thankful for the generous support of the computational resources by the Ohio Supercomputer Center.

ETHICS STATEMENT

We investigate a collaborative scenario in which higher-end detectors' predictions would ease other perception agents' labeling efforts in training their perception systems. In this paper, we focus on how to unleash the benefit of such a collaborative scenario, assuming that all the participating agents are benign. In practice, however, malicious agents may appear and degrade the overall pipeline. How to deal with malicious agents and data has been one of the main topics in AI with many robust solutions being proposed. We expect that these existing solutions can be incorporated into our work to mitigate potentially malicious situations.

REPRODUCIBILITY STATEMENT

We plan to make our implementation publicly available to promote reproducibility. Moreover, the implementation details, including all hyperparameters, model architectures, datasets, computational resources, and evaluation metrics, are provided in both the main paper (Sec. 4.1 Experimental Setups) and the supplementary material (Sec. S2 Additional Implementation Details).

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

# Supplementary Material for
# Learning 3D Perception from Others' Predictions

In this supplementary material, we provide implementation details and experimental results in addition to the main paper:

- Sec. S1: provides additional discussion of the scenario.
- Sec. S2: provides further implementation details.
- Sec. S3: provides additional empirical studies.

## S1   ADDITIONAL DISCUSSION

**More details on problem setup.** In our study, the overall goal is to develop a 3D perception model that can be deployed in an online setting. A conventional development process can typically be decomposed into four stages.

- Stage 1: data collection (online)
- Stage 2: data annotation, often by humans (offline)
- Stage 3: model training and validation (offline)
- Stage 4: model deployment and evaluation (online)

We exactly follows the four stages, except that in Stage 1, we assume that some nearby agents (*e.g.*, robotaxi, roadside unit) share their predictions as pseudo labels (*e.g.*, bounding boxes). We study how to leverage these pseudo labels to reduce human annotation in Stage 2 while maintaining the trained model quality in Stage 3.

Here, the stage where the ego car collects the nearby agents' predictions is the first stage, which is online. We note that there is no training or inference regarding the ego car's detector during this stage. After we collect the pseudo labels from the nearby agents, we refine the noisy pseudo labels with our proposed method and train the ego car's detector, which is offline (Stages 2 and 3). For our final model evaluation (Stage 4), which is online, the detector's computational cost is exactly the same as the standard detector.

**Discussion on offboard methods.** On the surface, both the offboard methods (Qi et al., 2021; Ma et al., 2023) and our learning scenario assume the existence of a pre-trained model. However, the accessibility to the model is different. More specifically, in the offboard methods, the pre-trained offboard model is directly accessible. One can use it to label the unlabeled data collected by the ego car. The resulting pseudo-labeled data can then be used to train the final onboard model. However, in our scenario, we do not have direct access to the pre-trained model, as it is deployed on the nearby agent, not the ego car. As such, we cannot use it to label the unlabeled data collected by the ego car. What we can access are the nearby agent's predictions on the data it collects, and we attempt to use them as pseudo labels of to train the ego car's onboard model.

## S2   ADDITIONAL IMPLEMENTATION DETAILS

The entire training pipeline for the experiments takes 2.5 hours with eight NVIDIA P100 GPUs. During training, we apply conventional data augmentation techniques such as rotation, scaling, and flipping following Xu et al. (2023). For training SECOND (Yan et al., 2018) in our ablation study, we set the number of epochs to 40, with the remaining training settings the same as PointPillars (Lang et al., 2019). For the domain adaptation experiments, we decrease the initial learning rate to 2e-4 and decay it to 2e-6, and fine-tune the model using the pre-trained parameters.

**Ranker architecture.** In the main paper, we train a ranker to select the best candidate from the sampled boxes. Specifically, we build our ranker upon PointNet (Qi et al., 2017), taking a normalized box and its corresponding points as inputs (see Fig. S1). We use two linear layers with ReLU

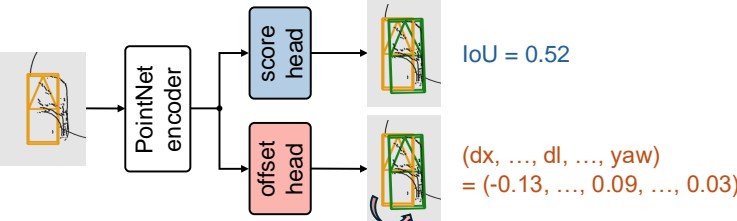

Figure S1: **Ranker architecture.** We give the initial noisy bounding box together with its nearby point clouds to the ranker as inputs and predict the IoU and offset to the ground-truth box.

non-linearity for both score head and offset head. Our box ranker's goal is similar to IoU scoring methods adopted in the existing detectors (Deng et al., 2021; Shi et al., 2020). However, we are motivated to build a ranker assuming we already have pseudo labels for objects but with a certain amount of localization errors. We demonstrate that this objective can be easily achieved with only a few frames (or even without any frame if we are able to use simulation data; Table 3 (a)) and without sophisticated architectural design. Therefore, we keep it to a very simple regressor that can already be satisfactorily used to demonstrate the effectiveness in refining the noisy pseudo labels.

**Ranker training.** We crop the point cloud to the range of $\times 3$ of the box size, and predict the IoU and the offset from the object. To prepare the training data, we take the first two frames for each of the 20 clips. We then sample approximately 100 boxes around each ground-truth label, crop out the point cloud around the sampled boxes to serve as training input, and compute the IoU and offset information for labels of the training set. During the ranker training, we also simulate random occlusion and point dropping. For the design of the loss function, we use a weighted combination: $\mathcal{L}_{\text{total}} = 5 * \mathcal{L}_{\text{IoU}} + \mathcal{L}_{\text{offset}}$. Here, the IoU loss $\mathcal{L}_{\text{IoU}}$ is defined by the mean squared error between $\hat{y}$ and $y$, where $\hat{y}$ denotes estimated IoU and $y$ denotes the actual IoU. For the offset prediction loss, since we only use the offset from the top-k boxes, the offset will most likely be applied to high IoU candidate boxes with smaller offset values, as shown in Fig. S2. As such, we are able to prevent training samples with lower IoU and larger offsets from dominating the training process. Based on this observation, we test several different loss functions and see that the ranker performed best using Smooth L1 loss, without adjusting for the sampled box's offset loss if the IoU is less than 0.3.

**Ranker inference.** For the ranker refinement module, we sample $N = 512$ boxes in total for both sampling strategies. In the naive sampling strategy, 512 boxes are sampled using Gaussian distributions with translational noise (on xyz) having a standard deviation of 1, and scaling and rotational noise (on lwh, yaw) having a standard deviation of 0.1, all with a mean of zero. For the C2F sampling, we employ both coarse and fine sampling. During coarse sampling, noise is uniformly sampled from $[-1.0, 1.0]$ for xy and from a Gaussian distribution with a standard deviation of 0.5 for z to help us sample $\frac{N}{2} = 256$ boxes. We then select the top $k = 3$ boxes based on the IoU reported by the ranker, apply the predicted offsets to these boxes, and proceed to sample a total of 256 boxes around each candidate for the fine sampling stage. In this stage, translational noise is sampled from a Gaussian distribution with a standard deviation of 0.25, noise for height and width with a standard deviation of 0.2, length with a standard deviation of 0.4, and rotational noise with a standard deviation of 0.1. We ultimately select the box with the highest IoU among all 256 sampled boxes and apply the predicted offset to obtain the refined label.

## S3    ADDITIONAL EMPIRICAL STUDIES

### S3.1    COMPARISON TO SEMI-SUPERVISED METHOD

We investigate a direct semi-supervised approach, using the 40 annotated frames and other unannotated frames of the ego car's data to train the detector. We apply 3DIoUMatch (Wang et al., 2021), a widely used and representative semi-supervised learning approach in this setting. We note that the official code of 3DIoUMatch used the PV-RCNN detector (Shi et al., 2020), not the PointPillar detector (Lang et al., 2019) in our main paper. As such, we rerun our approach using the PV-RCNN detector for a fair comparison.

Table S1: **Experiments on semi-supervised approach.** Our method and the semi-supervised method, 3DIoUMatch (Wang et al., 2021), can complement each other. The performance is reported on AP at IoU 0.7. * and † indicate supervision and approach, respectively.

| | labeled 40 frames* | $R$'s pred* | 3DIoUMatch† | Ranker (Sec. 3.3)† | Curriculum (Sec. 3.4)† | 0-30m | 30-50m | 50-80m | 0-80m |
|---|---|---|---|---|---|---|---|---|---|
| ① | ✓ | | ✓ | | | 60.7 | 24.4 | 6.0 | 37.1 |
| ② | ✓ | ✓ | | ✓ | ✓ | 65.1 | 34.1 | 12.6 | 41.7 |
| ③ | ✓ | ✓ | ✓ | ✓ | | 60.6 | 30.7 | 8.6 | 39.5 |
| ④ | ✓ | ✓ | ✓ | ✓ | ✓ | 68.1 | 36.8 | 12.9 | 44.4 |

Table S1 summarizes the results. ① is the result of 3DIoUMatch, and ② is the result of our approach (*c.f.*, Table 2 ⑩, but using PV-RCNN as the detector). We see that our approach outperforms 3DIoUMatch, demonstrating the value of using reference cars' predictions as auxiliary supervisions.

More importantly, we explore complementary nature of the two approaches. Specifically, we use our ranker to refine reference cars' predictions and add those high-quality ones (*i.e.*, < 40 meters, defined in Sec. 3.4) as extra labels to 3DIoUMatch. ③ shows the results: we see a 2.4% boost in 0-80 meters against ①, justifying the compatibility of ours and 3DIoUMatch. On top of ③, we further apply our distance-based curriculum for self-training (Sec. 3.4), using 3DIoUMatch's predictions on all the data as pseudo labels to re-train the detector. ④ shows the results: we see another 4.9% boost against ③ and 2.7% boost against ②. In sum, these results demonstrate 1) the effectiveness of our approach in leveraging reference cars' predictions as supervision (③ and ② *vs.*①) and 2) the compatibility of our approach with existing direct semi-supervised learning approaches to further boost the accuracy (④ *vs.*③ and ②). We view such compatibility as a notable strength: it demonstrates our approach as a valuable add-on when nearby agents' predictions are available.

## S3.2 EFFECT OF THE NUMBER OF TRAINING DATA

We conduct experiment to investigate how much the number of training data collected by following nearby agents could benefit the detector's final performance. In doing so, we train detectors with four different numbers of training clips, including 5, 10, 15, and 20, and report the overall AP at IoU of 0.5 and 0.7. The result in Table S2 shows that the performance consistently improves as the ego car collects more data (pseudo labels) from nearby agents. This again highlights the effectiveness of our newly explored scenario of learning from nearby agents' predictions.

Table S2: **Number of training data and performance.**

| | AP @ IoU | |
|---|---|---|
| # clips | 0.5 | 0.7 |
| 5 | 17.1 | 6.7 |
| 10 | 37.3 | 16.1 |
| 15 | 41.3 | 18.2 |
| 20 | 47.1 | 21.1 |

## S3.3 ANALYSIS ON SHARING DETECTOR

Our study introduces a novel learning scenario to build the detector by sharing object box predictions from the reference cars, which is realistic and practical. As shown in Table 1 in the main paper, the pseudo label quality achieved by R&B-POP is as competitive as directly sharing detector, which the scenario faces various constraints (*c.f.*, Sec. 1, Sec. 3.1). In this section, to investigate if our method can still improve such cases, we apply our algorithm on top of the object detector shared from the reference car. As shown in Table S3, we observe further performance

Table S3: **Detector performance on the ideal case.** R&B-POP also brings meaningful performance gain in the ideal scenario where the object detector can directly be shared.

| | AP @ IoU 0.7 | | | |
|---|---|---|---|---|
| method | 0-30m | 30-50m | 50-80m | 0-80m |
| sharing detector | 56.3 | 29.0 | 14.9 | 40.1 |
| + R&B-POP | 60.9 | 32.3 | 17.8 | 44.4 |

gain benefit from additional high-quality pseudo labels provided by the reference car in combination with our algorithm. We note that the overall performance is higher than Table 2 in the main paper, as the train and test sets share the same distribution (*i.e.*, clips 1-20 in Fig. 8 in the main paper). This highlights the effectiveness of our study of learning from reference agents' predictions beyond a single agent.

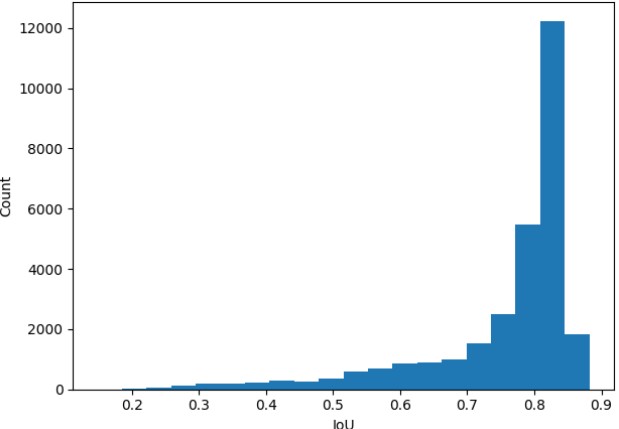

Figure S2: **Top IoU predicted from sampled boxes by ranker.** We computed the statistics for the IoU of the sampled boxes selected by the ranker during refinement. Then, the offset predicted by the ranker was applied to these selected boxes. The result indicates that the offset is most frequently used when the IoU of the box is sufficiently high.

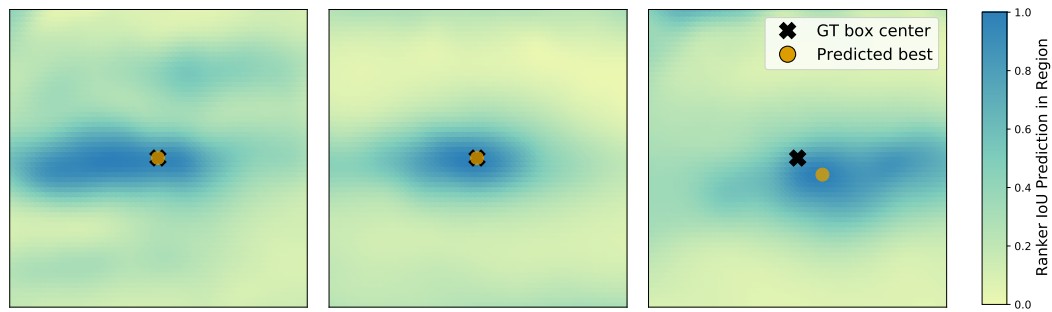

Figure S3: **Ranker IoU prediction behavior.** We visualize the IoU predicted by the ranker on sampled boxes versus the actual ground-truth location. We see that our ranker trained with a handful of annotated frames successfully refines initially mislocalized boxes by giving high scores to samples with more accurate box centers.

### S3.4 ADDITIONAL ANALYSIS ON RANKER

#### S3.4.1 VISUALIZATION OF PREDICTED SCORES

In the ranker training ablation study in our main manuscript, we mention our ranker design performs well with as little as 40 annotated frames. To illustrate its effectiveness further, we visualize the IoU prediction on sampled boxes *vs.* the actual location of the ground-truth as shown in Fig. S3. We observe that IoUs predicted by our ranker are consistent, and boxes with the highest predicted IoUs are close to the ground-truth. Moreover, the result implies that the ranker can effectively remove the false positives, as the region far away from the actual object tends to be given a lower score.

#### S3.4.2 RANKER ON HIGH DENSITY OF OBJECTS

To analyze the behavior of our ranker when objects are close, we carefully select examples where cars are close to each other (*i.e.*, within 2.5 meters of the box centers). We note that while the coarse sampling strategy considers translations of one to two meters, compared to the closest centers of two cars (*i.e.*, 2.5 meters), such translations do not necessarily misassign a box to a nearby car. We see that if the reference car predicts a box for each of the two nearby cars, our method can successfully recall both of them. Fig. S4 demonstrates the ranker's performance in correctly identifying and selecting the appropriate vehicles in the coarse sampling stage.

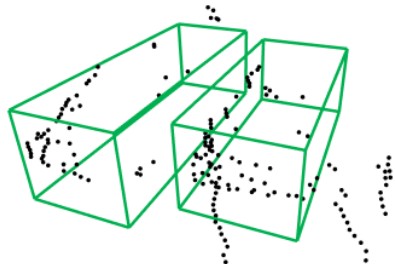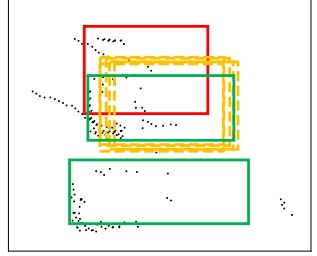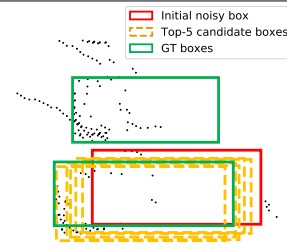

(a) Real example of a scenario where two objects with the **same category** are close.

(b) Visualization of the ranker refinement. Given an initial box assigned to a certain object, our ranker correctly identifies and selects the appropriate objects.

Figure S4: **Additional experiments on a high density of objects.**

Indeed, the purpose of the ranker is to refine the noisy pseudo labels from the reference to the correct location, size, and pose with respect to the ego agent's view (*c.f.*, Sec. 3.2). Therefore, even if the reference only predicts a single box or two predicted boxes are refined to the same car, leaving one false negative, our subsequent curriculum self-training is capable of discovering the remaining object.

### S3.4.3 EXTENSION TO MULTI-CLASS

In the main paper, we focus on a single category that includes various vehicles (*e.g.*, cars, trucks). Still, the pipeline proposed in the main paper is model-agnostic, meaning it can handle both single-class and multi-class detection. The key is to train separate rankers to capture class-specific information (*e.g.*, sizes, shapes) to provide high-quality pseudo labels for the subsequent distance-based curriculum self-training.

Table S4: **Extension of ranker to multi-class refinement.**

|  | car | truck |
|---|---|---|
| pseudo label | rec. / prec. | rec. / prec. |
| initial boxes | 56.1 / 71.0 | 57.2 / 61.0 |
| + our refinement | 60.6 / 76.7 | 64.2 / 68.4 |

Therefore, we explore the multi-class setup by further separating regular cars and trucks in V2V4Real (Xu et al., 2023) and employing car-specific and truck-specific rankers, respectively. As shown in Table S4, we observe significant improvements in label quality for both classes. Additionally, we conduct a study with selecting cases where nearby objects belong to different categories (*i.e.*, car *vs.* truck). This is considered challenging because cars and trucks have similar shapes but differ mainly in size. As shown in Fig. S5, we see that the two rankers capture class-specific information and score boxes of different sizes differently, for example, the car ranker gives smaller-size boxes a higher score. We believe such a property would reduce the chance of mistakenly assigning a box of one class to a nearby object of a different class. Moreover, by experimenting with tens of such cases with nearby objects of different classes, we find that the class-specific rankers can correctly maintain class distinctions (*i.e.*, not flipping the classes) with 72.5%, indicating that the rankers effectively capture class-specific information to provide high-quality pseudo labels.

### S3.5 ADDITIONAL ANALYSIS ON SELF-TRAINING

### S3.5.1 COMBINING DIFFERENT SETS OF PSEUDO LABELS

Table S5: **Anaylsis on pseudo label combination during self-training.** [†]distance-based curation: using only $R$'s predictions within its 40m.

| using $R$'s pred | distance-based curation[†] | AP @ IoU 0.5 | | | |
|---|---|---|---|---|---|
|  |  | 0-30m | 30-50m | 50-80m | 0-80m |
| x (main paper) | - | 73.3 | 43.3 | 23.3 | 56.5 |
| o | x | 71.8 | 41.5 | 26.5 | 55.1 |
| o | o | 74.5 | 42.0 | 25.1 | 57.0 |

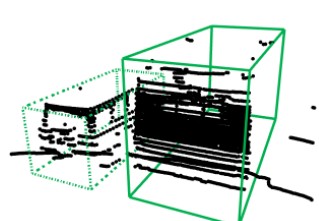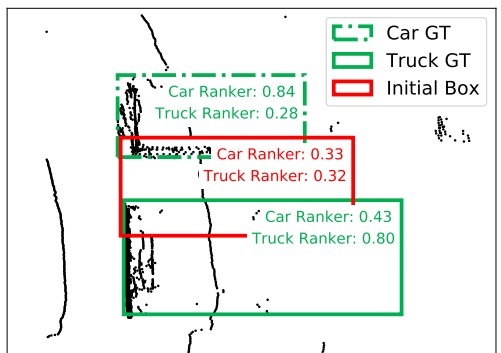

(a) Real example of a scenario where two objects with **different categories** ('car' and 'truck') are close.

(b) Visualization of two class-specific rankers (*i.e.*, car ranker and truck ranker). Each ranker assigns different scores to different bounding boxes but predicts the highest score for the desired class.

Figure S5: **Additional experiments on multi-class for ranker.**

In the main paper, we propose to utilze the predicted output of the trained detector, rather than the initially provided predictions from the reference car. In this section, we investigate different combining strategies. First, we naively combined the ego's and reference's predictions in Stage 2. As shown in Table S5, the overall performance (0-80m) dropped from 56.5 (row 1) to 55.1 (row 2). However, we also observe that the performance in the 50-80m range increased from 23.3 to 26.5. We hypothesize that predictions closer to the ego agent are actually farther from the reference, introducing noisier pseudo labels for the ego agent with the naive solution. Conversely, the reference provides more confident predictions for objects closer to it, which are farther from the ego-agent. Based on this assumption, we further explore a simple distance-based curation strategy, combining only predictions within 40m of the reference. As shown in the table, this approach improves the overall performance (0-80m) from 56.5 (row 1) to 57.0 (row 3) and maintained the performance in the 0-30m range (73.3 vs 74.5). These simple experiments demonstrate the potential for many interesting ideas that can be built upon our proposed learning scenario, and we leave it for the future study.

## S3.6    ADDITIONAL RESULTS ON OTHER DATASET

Table S6: **Additional experimental results on OPV2V dataset (Xu et al., 2022c).** The performance is reported on PointPillars (Lang et al., 2019) with 64-beam LiDAR. The evaluation metric is AP at IoU 0.5.  : uses GT labels.

|   | pseudo label | box refinement | self-training | time delay = 1 | | | | time delay = 2 | | | |
|---|---|---|---|---|---|---|---|---|---|---|---|
|   |   |   |   | 0-30m | 30-50m | 50-80m | 0-80m | 0-30m | 30-50m | 50-80m | 0-80m |
| ① | $R$'s pred | - | - | 84.4 | 62.7 | 31.1 | 71.7 | 80.9 | 58.6 | 22.0 | 67.3 |
| ② | $R$'s GT | - | - | 86.7 | 65.7 | 33.6 | 74.2 | 84.7 | 65.1 | 27.8 | 72.0 |
| ③ | $R$'s pred | ranker | - | 92.3 | 68.4 | 26.1 | 77.0 | 91.1 | 62.2 | 18.6 | 73.5 |
| ④ | $R$'s pred | - | distance-based curriculum | 94.4 | 74.7 | 28.2 | 80.8 | 94.1 | 72.8 | 28.0 | 79.9 |
| ⑤ | $R$'s pred | ranker | distance-based curriculum | 96.1 | 77.4 | 34.6 | 83.2 | 95.3 | 74.3 | 31.9 | 81.4 |
| ♣ | $E$'s GT | - | - | 97.6 | 89.4 | 68.4 | 90.8 | 97.6 | 89.4 | 68.4 | 90.8 |

In the main paper, we conduct experiments on the real-world dataset, V2V4Real (Xu et al., 2023). To see the generalizability of R&B-POP, we also evaluate our method on OPV2V (Xu et al., 2022c), a simulation dataset containing 2~7 connected cars within the scene. To suit our study, we re-split the entire 69 clips into 33 and 36 similar to Fig. 8 in the main paper. Also, we only use frames where the distance between the ego car and the reference car is within 90m. To simulate real-world noise, we sample random Gaussian noise with a zero mean and 0.2 standard deviation for localization error and consider a time delay of one and two frames. We set the training epoch to 15, and other hyperparameters remain the same. We use a total of 72 frames for the ranker training, which is two frames per ego car training clip. As shown in Table S6, we see that R&B-POP consistently improves performance on different data and settings, witnessing the general applicability of our method.

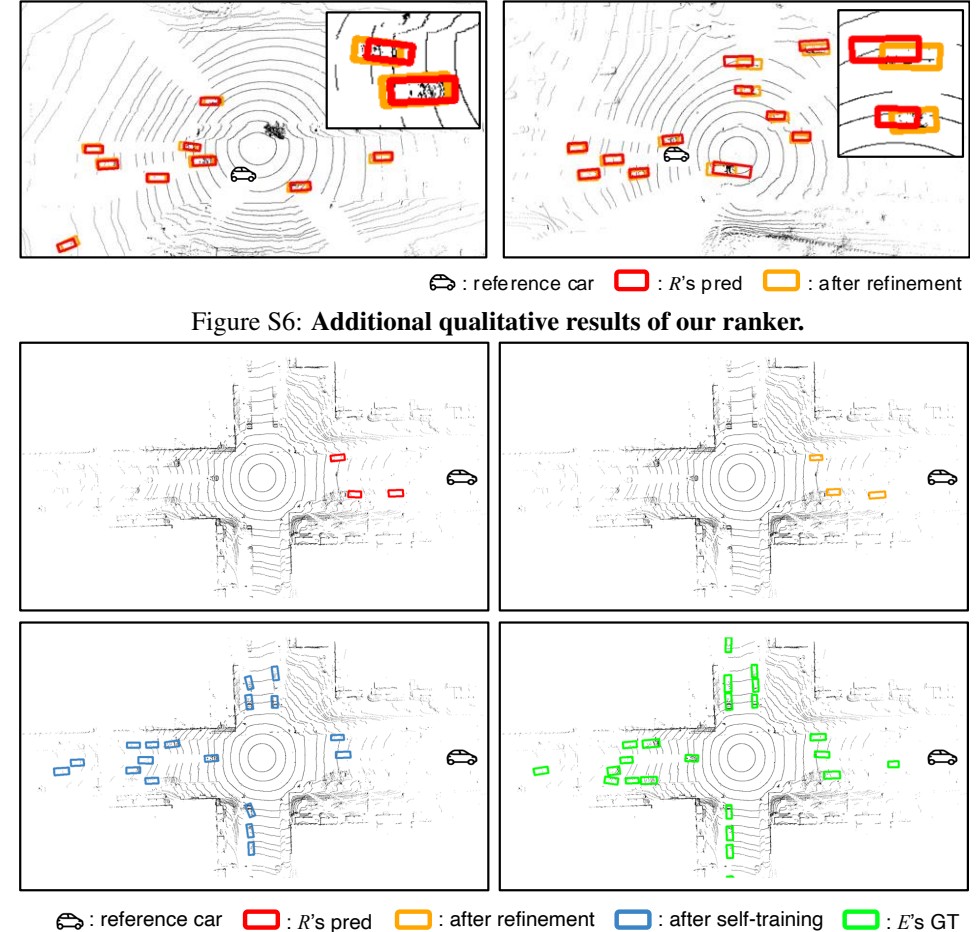

Figure S6: **Additional qualitative results of our ranker.**

Figure S7: **Additional qualitative results of our overall pipeline.**

### S3.6.1 MULTIPLE REFERENCE CARS

Table S7: **Experiments on the number of reference cars.** Detection performance improves further as the number of reference cars increases. The evaluation metric is AP at IoU 0.5.

| # reference car | time delay = 1 | | | | time delay = 2 | | | |
|---|---|---|---|---|---|---|---|---|
| | 0-30m | 30-50m | 50-80m | 0-80m | 0-30m | 30-50m | 50-80m | 0-80m |
| 1 | 96.1 | 77.4 | 34.6 | 83.2 | 95.3 | 74.3 | 31.9 | 81.4 |
| 2 | 94.3 | 78.2 | 43.5 | 83.4 | 96.1 | 79.1 | 38.6 | 84.0 |

Since OPV2V (Xu et al., 2022c) has scenes with more than one reference car, we investigate the relationship between the number of reference cars and detection performance. We use non-maximum suppression with a ranker score to combine two sets of pseudo labels. As shown in Table S7, we see that leveraging the predicted boxes from more reference cars improves final detection performance as different sets of pseudo labels from different views can supplement each other.

### S3.7 ADDITIONAL QUALITATIVE RESULTS

We provide additional visual results on V2V4Real (Xu et al., 2023) in Fig. S6 and Fig. S7. Notably, our pipeline improves pseudo label quality by adjusting mislocalization and by discovering and filtering out boxes appropriately.

