# OpenReview forum: "Learning 3D Perception from Others' Predictions"
_ICLR.cc/2025/Conference — ICLR 2025 Poster_

### Official Review · Reviewer_dJeW · 2024-10-22

**Soundness:** 3
**Presentation:** 3
**Contribution:** 3
**Rating:** 6
**Confidence:** 4

**Summary:**

This paper designs a new strategy to train the 3D object detector, i.e. learning from the detection results of a nearby unit equipped with a  pretrained detector. There are two main challenges, mis-localization and viewpoint mismatching, resulting in the inferior performance of naive self-training. To this end, this paper proposes a rank-based pseudo-label refinement module and a distance-based curriculum learning scheme. Experiments are conducted on V2V4Real and OPV2V datasets to evaluate the proposed algorithm.

**Strengths:**

1. This paper proposes a creative and interesting strategy for the training of object detectors. It can exploit existing pretrained detector in a flexible way regardless of their input formats and deployment compatibility. The authors provide detailed explanation about their motivation and sound justification of the feasibility.

2. The algorithm includes a relatively complex pipeline. However, each module inside it is well-motivated and has a clear role. The step-by-step improvement brought by each module is also demonstrated clearly through extensive experiments.

3. The paper is well-written. The authors organize everything well to make it easy to follow. The illustrations are also very clear, helpful for  readers to understand the content.

**Weaknesses:**

1. The authors are targeted for LiDAR sensors. However, if we apply this setting in practice, the camera sensors are much more ubiquitous such as the surveillance. I am wondering whether the algorithm would still work if the reference car is using camera-based detector. Given camera-based detectors are weaker, this may require high robustness of the proposed algorithm.

2. In the experiments, it may be helpful to include some baselines using other strategies for the training of detector. For example, since the algorithm requires a small number of annotated frames to train the ranker, it may alternatively apply some semi-supervised strategies to train the detector directly.

**Questions:**

Please consider replying to the Weaknesses.

---

> ### Author Response · Authors · 2024-11-20
> **Response to Reviewer dJeW**
>
> We thank your valuable feedback to improve our manuscript further. Below, we respond to most, if not all, of your comments at this stage.
>
> ## Answer to Weakness 1: Reference agents with camera-based detector.
>
> Thank you for pointing this out! We are indeed glad to hear our study has great potential to lead to several promising research directions. We want to humbly highlight that this is the first paper to explore such a scenario to our knowledge, and we hope that our findings and approach have set the stepping stone to future research in this direction, including reference agents with camera-based detectors.
>
> While it will be interesting to study the case where reference agents are equipped with camera-based detectors, we find it not to be easy and straightforward to directly conduct the study. Firstly, the dataset that we have used, V2V4Real, doesn’t contain camera frames. Moreover, as the reviewer mentioned, a lack of depth information in camera-based detectors could potentially lead to unsatisfactory performance for reference agents, and the ego car’s detector could result in suboptimal performance if we leverage predictions from such relatively less accurate predictions. However, we also want to note that reference agents do not necessarily need to be general experts, but could be local experts (Sec 3.1. Feasibility and practicality; Lines 174 - 179) in the geo-fenced area, and it could lessen the level of difficulty for reference agents to be well-trained. Still, we believe this challenge could be an interesting research question, and we leave it for future study.
>
> Finally, we humbly note that no matter what sensors and detectors (camera-based or LiDAR-based) are in use, if we aim at 3D perception, they will generate 3D bounding boxes (as pseudo labels). We believe that those pseudo labels will likely show inherent localization, synchronization, and calibration errors, and the fundamentals of findings and solutions in our study would still hold.
>
> ## Answer to Weakness 2: More baseline training methods.
>
> Thank you for your comment. Based on your feedback, we are currently conducting experiments regarding the existing semi-supervised method, but it’s a bit delayed due to resource limits. We will share our observations when we have the results.
>
> Again, we appreciate your valuable opinions, which will significantly improve our study. Please also kindly give us a chance to clarify any remaining unclear points.

---

> > ### Author Response · Authors · 2024-11-27
> >
> > We thank the reviewer again for your invaluable comments. Based on your feedback, we have revised our manuscript. Please don’t hesitate to let us know if you have any remaining concerns. Additionally, **we are currently conducting experiments to compare with a semi-supervised method**, as suggested. We anticipate sharing the results soon and sincerely appreciate your patience!

---

> > > ### Author Response · Authors · 2024-12-03
> > >
> > > Dear Reviewer dJeW,
> > >
> > > We would like to give you some updates:
> > > - First, due to some computational issues, we need to delay a little bit about semi-supervised learning experiments, but we will surely try our best to provide the result by the end of the discussion period.
> > > - Second, we appreciate your positive opinion of our paper. In light of our further clarification and experimental result, we want to kindly ask if there is an additional response you would like us to provide by December 3rd so that you may consider further increasing your score.

---

> ### Comment · Reviewer_dJeW · 2024-12-03
>
> I appreciate the response from authors. I was waiting for the semi-supervised learning results, so I did not reply until the last day.
>
> For the first concern, I can understand that it is the constraint of the dataset. However, I think my second concern is more important. Semi-supervised learning should be a crucial baseline for this paper.
>
> Besides, I partially agree with Reviewer R3jy in the application of off-board autolabelling. From my perspective, offboard auto-labeling models, which can also generate pseudo-labels, should be more accessible than those reference detector results on the street from different organizations. People can use these offboard models to annotate data and then train other online models. I understand this paper works on a different scenario and acknowledge the novelty of this scenario, so I do not require the authors to beat those offboard autolabling baselines (I believe it is difficult). However, it to some extent still hurts the practical usefulness of this proposed method.

---

> > ### Author Response · Authors · 2024-12-03
> > **Response by Aythors**
> >
> > Dear Reviewer dJeW,
> >
> > We appreciate your timely feedback.
> >
> > Since there is only one more hour to discuss with you and receive your further feedback, please pardon us for a quick, unpolished response. We will provide a more detailed response by the Dec 3rd deadline.
> >
> > First, we apologize for the delay in the semi-supervised learning results. The computational issue was not expected.
> >
> > Second, we appreciate your comments on semi-supervised learning as crucial baselines. As will be shown, our method that leverages reference car's predictions as additional supervision could outperform conventional semi-supervised learning methods that solely rely on the limited ground-truth labels (in our setting, 40 frames). That said, we certainly do not claim that our approach is "more superior." Our results are to demonstrate the value of leveraging nearby agents' predictions. Indeed, as will be shown, our method and conventional semi-supervised learning are compatible and can be integrated to boost the performance further. (Please also see our response to R3jy "Answer to Weakness 3: Naive approach vs. semi-supervised learning" for further discussions.)
> >
> > Third, we appreciate your comments on the offboard setting. In our humble opinion, the offboard setting and our setting are not competitors, but two ways to leverage specific data cues to improve label efficiency. In an offboard setting, one can use the future frame information (in the next few seconds) to improve labeling. In our setting, one can leverage the predictions by other nearby agents to reduce human annotation. As such, they can, in theory, be combined (we will discuss this point further). Please also see our response to R3jy "Thank you for your timely response [1/2]" and "Thank you for your timely response [2/2]" for a detailed discussion.
> >
> > In sum, we respectfully think what makes auto-driving an interesting research domain is its rich cues inside the data. Offboard settings and some other settings (e.g., using data along repeated routes) investigate some of the cues, and our paper explores a new one. We will provide more discussions by the Dec 3rd deadline.

---

> > > ### Author Response · Authors · 2024-12-04
> > >
> > > We again thank you for your thoughtful comments. We have posted a new experiment regarding semi-supervised learning in the general response. Please kindly refer to our response. To summarize:
> > >
> > > - Our method outperformed a representative semi-supervised method, demonstrating the value of using reference cars’ predictions as auxiliary supervision.
> > > - More importantly, our method and the semi-supervised method leverage complementary data cues and can be combined to boost the performance further.
> > >
> > > We humbly believe that our response has adequately highlighted the value of the new learning scenario we explore. We also respectfully think that we have addressed most, if not all, of the concerns raised in the author-reviewer discussion period. We appreciate your consideration and being supportive of our study.

---

### Official Review · Reviewer_goE2 · 2024-10-31

**Soundness:** 3
**Presentation:** 4
**Contribution:** 3
**Rating:** 6
**Confidence:** 4

**Summary:**

This papers proposes the novel idea of "learning from others predictions", i.e. using pseudo-labels from a detector deployed on another agent in the scene to train an ego agent perception model. The approach can be summarized in three parts: first, the reference detector is pre-trained on a human-annotated labels offline, and a box ranker is pre-trained on a small number of human-annotated ego frames. Next, the reference detector's pseudo-labels are generated for frames in which the reference agent is below some threshold distance from the ego agent, and are used to train the ego detector after undergoing a course-to-fine box refinement module with the box ranker. Lastly, self-training is performed using the ego detector's predictions on all available frames. Their approach, dubbed R&B-POP, is able to significantly improve AP from naively using the reference detector predictions, nearly equaling performance of training with human-annotated labels on V2V4Real Collaborative Perception dataset.

**Strengths:**

1. The setting introduced in this paper is novel and creative, and will inspire future work in this new research direction.
2. The method is well-explained and intuitive. Using a set of thoughtful approaches for improving psuedo-label quality and training, the authors are able to significantly improve performance over the baseline and other heuristic approaches.
3. Experiments are very thorough, and many ablation studies have been performed.
4. The paper is well-written and organized.

**Weaknesses:**

1. Although the results are impressive on V2V4Real, the experiments are performed on the case where there is minimal domain shift between the two vehicle detectors (e.g. they both have the same sensors, both survey the scene from the street, etc). It would be a more interesting experimental setting with two drastically different object detectors, e.g. one on a LiDAR mounted car, and one on a stationary security camera. The authors suggest scenarios such as these as possible applications, although it's not clear that the method demonstrated in this work can be easily applied to such a scenario. However, I understand experiments are limited by available data, so these may not be feasible in practice.
2. The authors utilize a small number of human-annotated frames from the Ego vehicle sensor data to train the box ranker, so the need for a human labeler is not completely eliminated. It would also be compelling to include experiments comparing to other semi-supervised learning approaches for 3D object detection using a similar number of labeled ego frames, to demonstrate that R&B-POP works better than conventional SSL approaches for this level of annotator budget.

**Questions:**

Have the authors tried training the ranker without using any annotated ego frames? As mentioned previously, it would be compelling if the need to label any of the ego data was completely eliminated.

---

> ### Author Response · Authors · 2024-11-20
> **Response to Reviewer goE2**
>
> We thank your valuable feedback to improve our manuscript further. Below, we respond to most, if not all, of your comments at this stage.
>
> ## Answer to Weakness 1:  About agents with different modalities.
>
> Thank you for pointing this out! We are indeed glad the reviewer found our study to have great potential to encourage several promising research directions. We would humbly like to highlight that this is the first paper to explore such a scenario to our knowledge, and we hope that our findings and approach have set the stepping stone to future research in this direction.
>
> We would also like to note that while it would be interesting to study such extreme cases (*e.g.*, drastically different object detectors), it would not be easy and straightforward at the same time. Firstly, the dataset that we have used, V2V4Real, doesn’t contain camera frames. Moreover, if we consider camera-based reference agents and LiDAR-based ego car, camera-based detectors would typically perform worse than LiDAR-based detectors because of a lack of depth information. If we leverage predictions from such relatively less accurate predictions, the ego car’s detector could result in suboptimal performance. While we have provided a discussion about lessening the level of difficulty for reference agents to be well-trained (*e.g.*, leveraging the repetition or background cues; Lines 174 - 179), we still believe this challenge could be one of the interesting research questions to explore. The opposite case where LiDAR-based reference agents and camera-based ego car would also be promising, and we leave them for future study.
>
> Finally, we humbly believe that no matter what sensors and detectors (camera-based or LiDAR-based) are in use, if we aim at 3D perception, they will generate 3D bounding boxes (as pseudo labels). We believe that those pseudo labels will likely show inherent localization, synchronization, and calibration errors, and the fundamentals of findings and solutions in our study would still hold.
>
> ## Answer to Weakness 2 and Question: About fully unsupervised setting.
>
> We have already investigated a fully unsupervised setting by training our ranker with only synthetic data (Table 3 (a) in the main paper). The result shows the potential for entirely replacing labeling effort if we are able to access simulation data (Lines 464 - 470). Still, at the same time, we are currently conducting experiments regarding the existing semi-supervised method in response to the feedback, but it’s a bit delayed due to resource limits. We will share our observations when we have the results.
>
> Again, we appreciate your valuable opinions, which will significantly improve our study. Please also kindly give us a chance to clarify any remaining unclear points.

---

> > ### Comment · Reviewer_goE2 · 2024-11-27
> >
> > Thank you for the response. I am satisfied with the response to my concerns, however I still stand by my previous observations so I keep my score unchanged.

---

> > > ### Author Response · Authors · 2024-11-27
> > >
> > > We thank the reviewer again for your timely response and invaluable comments. Based on your feedback, we have revised our manuscript. Please don’t hesitate to let us know if you have any remaining concerns. Additionally, **we are currently conducting experiments to compare with a semi-supervised method**, as suggested. We anticipate sharing the results soon and sincerely appreciate your patience!

---

> > > > ### Author Response · Authors · 2024-12-03
> > > >
> > > > Dear Reviewer goE2,
> > > >
> > > > We would like to give you some updates:
> > > > - First, due to some computational issues, we need to delay a little bit about semi-supervised learning experiments, but we will surely try our best to provide the result by the end of the discussion period.
> > > > - Second, we appreciate your positive opinion of our paper. In light of our further clarification and experimental result, we want to kindly ask if there is an additional response you would like us to provide by December 3rd so that you may consider further increasing your score.

---

### Official Review · Reviewer_LF3v · 2024-11-02

**Soundness:** 3
**Presentation:** 3
**Contribution:** 2
**Rating:** 6
**Confidence:** 3

**Summary:**

The paper investigates a new scenario for constructing 3D object detectors in autonomous driving by leveraging predictions from nearby units equipped with accurate detectors. The paper addresses the challenges of viewpoint mismatches and mislocalization due to synchronization and GPS errors. The authors propose a learning pipeline called Refining & Discovering Boxes for 3D Perception from Others' Predictions (R&B-POP). This pipeline includes a distance-based curriculum that starts by learning from closer units with similar viewpoints and then improves the quality of predictions from other units through self-training. In general, while there are no surprising design elements in the methodology, the problem addressed in the paper is a new issue in this field. Learning new knowledge from the predictions of other agents appears to have significant practical application value.

**Strengths:**

1. The paper is well-structured, with a clear abstract, introduction, methodology, experiments, and conclusion sections that logically flow from one to the next.
2. The proposed  Refining & Discovering Boxes for 3D Perception from Others’ Predictions (R&B-POP) method is simple and effective.
3. The paper has extensive experiments and the figures are very professionally made.

**Weaknesses:**

1. The scenario assumed in the paper has great limitations, that is, learning knowledge from nearby agents to achieve detection. Theoretically, the learning process needs to be fast enough so that the current vehicles can predict new objects. However, I did not see the authors' analysis of the algorithm's training speed and discussion of actual deployment limitations.
2. Box ranker seems to be just an IoU scoring method, which is adopted in most detectors, such as Voxel-RCNN, PV-RCNN, they are not fundamentally different, why not directly use or discuss the limitations of existing IoU scoring method in this application?

**Questions:**

1. It is possible that the method can learn knowledge from other vehicles in real-time, which would expand the scope of practical applications?
2. What's the key difference between Box ranker and IoU scoring in traditional detectors?

---

> ### Author Response · Authors · 2024-11-20
> **Response to Reviewer LF3v**
>
> We thank your valuable feedback to improve our manuscript further. Below, we respond to most, if not all, of your comments at this stage.
>
> ## Answer to Weakness 1: Clarification of our setting.
>
> We apologize for any unclear writing that could potentially lead to a misunderstanding of the scenario we focus on. Our overall goal is to develop a 3D perception model that can be deployed in an online setting. A conventional development process can typically be decomposed into four stages.
>
> - Stage 1: data collection (online)
> - Stage 2: data annotation, often by humans (offline)
> - Stage 3: model training and validation (offline)
> - Stage 4: model deployment and evaluation (online)
>
> Our paper exactly follows the four stages, except that in Stage 1, we assume that some nearby agents (*e.g.*, robotaxi, roadside unit) share their predictions as pseudo labels (*e.g.*, bounding boxes). Our main contribution is in how to leverage these pseudo labels to reduce human annotation in Stage 2 while maintaining the trained model quality in Stage 3. We will rewrite our problem setup to clarify this (Lines 157 - 161).
>
> Here, the stage where the ego car collects the nearby agents’ predictions is the first stage, which is online. We note that there is no training or inference regarding the ego car’s detector during this stage. After we collect the pseudo labels from the nearby agents, we refine the noisy pseudo labels with our proposed method and train the ego car’s detector, which is offline (Stages 2 and 3). We humbly note that reducing the computational cost during these stages (*e.g.*, resource-efficient training or inference) is not the main goal of this study. For our final model evaluation (Stage 4), which is online, the detector’s computational cost is exactly the same as the standard detector.
>
> **Q: It is possible that the method can learn knowledge from other vehicles in real-time, which would expand the scope of practical applications?**
>
> We are glad that the reviewer found our scenario has the potential to raise several interesting topics. However, we respectfully want to emphasize that this is the first study to explore a new research direction, and we didn’t consider such a setting in this study. However, we humbly believe learning from the nearby agents in an online manner would be a very promising and practical direction.
>
> ## Answer to Weakness 2 and Question 2: Box Ranker vs. existing IoU scoring methods.
>
> Thank you for pointing out. We agree that our box ranker’s goal is similar to IoU scoring methods adopted in the existing detectors, such as Voxel-RCNN and PV-RCNN. We respectfully think that the key difference is in its motivation and usage. In our study, we are motivated to build a ranker assuming we already have pseudo labels for objects but with a certain amount of localization errors. Therefore, the objective of our ranker is to find the best candidate among a bunch of candidate boxes near the initial noisy pseudo label so that the selected box can serve as better pseudo labels. In our study, we wanted to demonstrate that this objective can be easily achieved with only a few frames (or even without any frame if we are able to use simulation data; Table 3 (a) in the main paper) and without sophisticated architectural design. While we agree that our ranker could be replaced with the existing methods, we kept it to a very simple regressor that can already be satisfactorily used to demonstrate the effectiveness in refining the noisy pseudo labels. To this end, we choose to use a simple pointnet-based model. We will add the discussion to our manuscript.
>
> Again, we appreciate your valuable opinions on improving our study. Please also kindly give us a chance to clarify any remaining unclear points.

---

> > ### Comment · Reviewer_LF3v · 2024-11-22
> > **Response to Author**
> >
> > I have carefully read all comments and the author's responses.  I think that although the techniques used by the authors are mostly previous semi-supervised techniques, learning new knowledge from the prediction results of other agents is a very new problem. In particular, I look forward to seeing an online method in the future that can learn from other agents in real-time to expand their perception. Anyway, I keep my original recommendation.

---

> > > ### Author Response · Authors · 2024-11-27
> > >
> > > Thank you for recognizing the value of our study and for your positive outlook on accepting our paper. We are also excited about the potential to explore learning from nearby agents in an online manner in the future. We humbly believe this study serves as a solid stepping stone for this new learning scenario. Please do not hesitate to let us know if you have any remaining concerns –  we are more than happy to address them!

---

> > > > ### Author Response · Authors · 2024-12-03
> > > >
> > > > Dear Reviewer LF3v,
> > > >
> > > > We would like to give you some updates:
> > > > - First, due to some computational issues, we need to delay a little bit about semi-supervised learning experiments, but we will surely try our best to provide the result by the end of the discussion period.
> > > > - Second, we appreciate your positive opinion of our paper. In light of our further clarification and experimental result, we want to kindly ask if there is an additional response you would like us to provide by December 3rd so that you may consider further increasing your score.

---

### Official Review · Reviewer_iYjB · 2024-11-02

**Soundness:** 2
**Presentation:** 3
**Contribution:** 2
**Rating:** 6
**Confidence:** 2

**Summary:**

This paper aims to develop a label-efficient approach for training 3D object detectors in autonomous vehicles by leveraging predictions from nearby vehicles equipped with accurate detectors. The main technical contribution is the proposed R&B-POP pipeline, which refines detection results from neighboring cars using a box ranker and a distance-based curriculum self-training approach. This method addresses challenges like viewpoint mismatches and localization errors that arise when integrating predictions from external sources. In experiments, the authors validated their approach on a collaborative driving dataset; they showed that with a small set of labeled data, this approach achieves performance close to that of fully supervised models.

This is an interesting exploration of new formulation for label-efficient 3D object detection. However, it is unclear how well the proposed method performs compared to other data-efficient domain adaptation or fine-tuning strategies, as relevant comparisons with SOTA are lacking. Additionally, the problem setting feels somewhat artificial, relying on several assumptions that may limit its practical applicability.
My rating is a weak reject.

**Strengths:**

+ The paper introduces an interesting problem formulation for 3D object detection in resource-limited settings, suggesting a new collaborative information-sharing approach to reduce labeling costs.
+ The ablation study is well designed. Results demonstrate the contribution of different design components to the overall performance, sharing valuable insights into the effectiveness of these different components in the proposed pipeline.

**Weaknesses:**

-	Practicality and applicability of the problem setting. One of my main concerns is the practicality of the proposed problem setting. First, the approach relies heavily on the assumption that neighboring cars with accurate detectors are always available when needed, but this may not always be feasible. Additionally, without having a good pre-trained detector but attempting to rely on other cars’ detection sharing seems unsafe. This mechanism will also need to resolve multiple other dependencies and complexities such as identifying trustworthy vehicles (e.g., not malicious, not inaccurate), managing potential error accumulation (causing problems when passing to other cars), etc. Without all these sorted out, the current problem setting is somewhat artificial and lacks reliability and scalability.
-	Technical comparisons. If the primary technical goal is to enable fine-tuning with minimal data, it should include comparisons with existing/ SOTA data-efficient training and fine-tuning methods. Demonstrating how this method compares with these techniques/models, will better clarify its advantages and limitations. Such a comparison is currently lacking.
-	Another minor issue: when comparing with the fully supervised models, please also include comparisons with SOTA models. Additionally, consider testing how the accuracy changes if fewer data are selected/used for training. This comparison would better illustrate the proposed pipeline’s advantage over simply having a road-side transmitter providing local data to passing vehicles.

**Questions:**

please see my major concerns in the weaknesses section.

---

> ### Author Response · Authors · 2024-11-20
> **Response to Reviewer iYjB [1/2]**
>
> Thank you for the valuable feedback and we apologize for any unclear explanation that has led to misunderstanding. Below, we respond to most, if not all, of your comments at this stage.
>
> ## Answer to weakness 1: Practicality and applicability of the problem set.
>
> Thank you for your detailed comment. In our current version, we have made some effort to justify the practicality and applicability. Please kindly refer to Lines 46 - 51 and Lines 174 - 191. We hope these text portions address your concerns to a certain extent. In the following, we provide specific responses.
>
> **Q. First, the approach relies heavily on the assumption that neighboring cars with accurate detectors are always available when needed, but this may not always be feasible.**
>
> We note that we rely on nearby agents’ predictions only during the data collection phase. We provide a decomposition of the model development process in response to Reviewer R3jy (Answer to Weakness 1. Stage 1 - stage 4), in which the data collection is in Stage 1. In other words, as long as nearby cars were present during part of the data collection, we can already leverage their predictions to train the model; the other portion of the data can be leveraged in a self-training fashion (*cf.* Step 2 in Section 3.5).
>
> We also want to note that nearby cars are not the only sources of pseudo-label providers. Static agents like roadside units can also offer pseudo labels. These agents can be more easily deployed to cover a wide range of drivable areas, and their detectors can be more easily trained (please refer to Lines 174 - 179).
>
> Last but not least, collaborative autonomous driving (CAD) has indeed become a promising framework to investigate in the self-driving, smart city, and transportation community. It is reasonable to expect that nearby agents’ predictions will be easier to obtain over time. As a research paper, we respectfully think that building a stepping stone for future scenarios should be encouraged.
>
> **Q. … attempting to rely on other cars’ detection sharing seems unsafe … This mechanism will also need to resolve multiple other dependencies and complexities such as identifying trustworthy vehicles (e.g., not malicious, not inaccurate), managing potential error accumulation (causing problems when passing to other cars), etc.**
>
> Building upon our response above, we would like to emphasize that collaborative autonomous driving (CAD) is a multi-faceted, cross-disciplinary topic. It is not a pure computer vision problem; instead, many papers within the scope study problems involving communication, infrastructure, security, and privacy ([a, b, c, d, e, f]). It is indeed encouraging as it allows the vision community to focus on developing collaborative perception methods to leverage data with mild synchronization and mislocalization errors [g, h, i, j as well references in Lines 142 - 144] while leaving some other concerns (e.g., communication safety and trustworthiness) to the research communities with expertise. In our paper, we follow most of the existing collaborative perception papers to assume the transmitted data does not include malicious data. We will include this additional discussion.

---

> > ### Author Response · Authors · 2024-11-20
> > **Response to Reviewer iYjB [2/2]**
> >
> > ## Answer to weakness 2: Technical comparisons.
> >
> > We would like to emphasize that our main goal is not to fine-tune with minimal data but to explore the opportunities to leverage other supervision to ease the annotation effort. As such, our method is not a competitor to existing label-efficient approaches (see related work), but a complementary approach to them. For example, our method incorporates self-training (a semi-supervised approach) to better leverage nearby agents’ noisy predictions (Section 3.4); we can also incorporate parameter-efficient fine-tuning approaches, if needed, to speed up fine-tuning and prevent over-fitting.
> >
> > ## Answer to weakness 3: Another minor issue.
> >
> > **Comparison with SOTA.**
> >
> > We respectfully note that our choice of detector models (*i.e.*, PointPillar) follows the original V2V4Real dataset paper [k]. Our main focus is to show that we can train the model without extensively annotated data by effectively leveraging other agents’ predictions as pseudo-labels, not to compare to SOTA detectors. Our method is detector-agnostic, and, in theory, can be applied to any detector. We have indeed conducted such an analysis in Table 5 (b) and Lines 510 - 514.
> >
> > **# data and performance.**
> >
> > Thank you for your insightful comment! We are currently experimenting in response to your suggestion and will add our findings shortly.
> >
> > **Seeking feedback.**
> >
> > Again, we appreciate your valuable opinions, which will significantly improve our study. Please also kindly give us a chance to clarify any remaining points that are unclear.
> >
> > ----
> > [a] Feng et al., “ Vulnerability of traffic control system under cyberattacks with falsified data,” Transportation Research Record, 2018.
> >
> > [b] Chen et al., “Exposing congestion attack on emerging connected vehicle-based traffic signal control,” Network and Distributed System Security (NDSS) Symposium, 2018.
> >
> > [c] Lu et al., "Performance specifications and metrics for adaptive real-time systems," IEEE Real-Time Systems Symposium, 2000.
> >
> > [d] Seif and Hu, “Autonomous driving in the iCity—HD maps as a key challenge of the automotive industry,” Engineering, 2016.
> >
> > [e] Mao et al., “A survey on mobile edge computing: The communication perspective,” IEEE
> > Communications Surveys & Tutorials, 2017.
> >
> > [f] Wang et al., “Tracking hit-and-run vehicle with sparse video surveillance cameras and mobile taxicabs,” in ICDM, 2017.
> >
> > [g] Lei et al., “Latency-aware collaborative perception,” in ECCV 2022.
> >
> > [h] Wei et al., “Asynchrony-robust collaborative perception via bird's eye view flow,” in NeurIPS 2024.
> >
> > [i] Lu et al., “Robust collaborative 3d object detection in presence of pose errors,” in ICRA 2023.
> >
> > [j] Vadivelu et al., “Learning to communicate and correct pose errors,” in CoRL 2021.
> >
> > [k] Xu et al., “V2v4real: A real-world large-scale dataset for vehicle-to-vehicle cooperative perception,” in CVPR 2023.

---

> ### Author Response · Authors · 2024-11-21
> **Additional experiment**
>
> **Additional experiment on # data and performance.**
>
> In response to the reviewer’s suggestion, we have conducted an additional experiment to investigate how much the number of training data collected by following nearby agents could benefit the detector’s final performance. In doing so, we train detectors with four different numbers of training clips, including 5, 10, 15, and 20, and report the overall AP at IoU of 0.5 and 0.7.
>
> | # training clips   | AP @ 0.5 |  AP @ 0.7 |
> |----------------------|:--------:|:---------:|
> | 5                    |   30.6   |    11.8   |
> | 10                   |   49.0   |    26.1   |
> | 15                   |   55.0   |    28.8   |
> | 20 (our final model) |   56.5   |    32.6   |
>
> The above table shows that the performance consistently improves as the ego car collects more data (pseudo labels) from nearby agents. We humbly believe this again highlights the effectiveness of our newly explored scenario of learning from nearby agents’ predictions. We thank the reviewer for suggesting this helpful experiment and will add the result to our manuscript.
>
> Again, please kindly give us a chance to clarify any remaining points that are unclear.

---

> ### Author Response · Authors · 2024-11-27
>
> We sincerely thank the reviewer for taking time to review our rebuttal and for being willing to increase the score. We also thank you for highlighting our study as a valuable exploration. We humbly believe that our study of this new learning scenario has the potential to open a new direction and provide meaningful insights to the community. Please let us know if you have any additional concerns or suggestions – we would be more than happy to address them and engage in further discussion!

---

> ### Author Response · Authors · 2024-12-03
>
> Dear Reviewer iYjB,
>
> We would like to give you some updates:
> - First, due to some computational issues, we need to delay a little bit about semi-supervised learning experiments, but we will surely try our best to provide the result by the end of the discussion period.
> - Second, we appreciate your positive opinion of our paper. In light of our further clarification and experimental result, we want to kindly ask if there is an additional response you would like us to provide by December 3rd so that you may consider further increasing your score.

---

### Official Review · Reviewer_R3jy · 2024-11-03

**Soundness:** 2
**Presentation:** 3
**Contribution:** 2
**Rating:** 5
**Confidence:** 5

**Summary:**

This paper introduces a novel approach for training 3D object detectors by utilizing predictions from nearby agents with accurate detectors. This approach addresses the challenges of acquiring large amounts of annotated data. Validated on a real-world collaborative driving dataset, the proposed method demonstrates effective label-efficient learning and substantially improves the AP.

**Strengths:**

- The quality of the paper's figures is high and relatively clear and explicit.
- This method ensures the accuracy of the detector while reducing the cost of annotation, and its effectiveness has been verified on both real and simulated datasets.

**Weaknesses:**

1. This paper introduces a variant of the offboard 3D object detection method. It only selects an unsupervised scheme as a baseline, which is inappropriate. For a fair comparison, the proposed paper should be compared with offboard 3D object detection methods with similar experimental settings. These methods also rely on an accurate detector to label unannotated scenes. The authors should select the correct baselines[1] [2] for comparison.
[1] DetZero: Rethinking Offboard 3D Object Detection with Long-term Sequential Point Clouds, ICCV 2023.
[2] Offboard 3D Object Detection from Point Cloud Sequences. CVPR 2021.

2. Compared to traditional offboard 3D object detection, the proposed solution not only uses full data annotation to train an accurate detector but also provides partial manual labeling for unannotated scenarios. This approach of adding extra manual labeling diminishes the contribution of the method. The authors emphasize that when an autonomous vehicle enters a new scene, it can obtain scene information from an established intelligent agent, which is an online process. However, the method itself is offline, which creates a contradiction. What are the proposed method's contributions in light of these concerns?
3. The proposed method provides pseudo-labels for the ego vehicle by directly passing the predicted results of the reference vehicle, which is a very naive approach. A semi-supervised learning framework is more flexible. What kind of performance effects would a semi-supervised learning framework have in this experimental setup? For example, train a teacher network using data from labeled reference vehicles, and then use the teacher network to generate pseudo-labels for unlabeled point clouds to train the student network further.
4. The ground truth of the reference vehicle is completely discarded in the subsequent method design, which is undoubtedly regrettable. Even if the center position of the ground truth has a deviation for the ego vehicle, the size of the bounding box is entirely correct, and this information is also very important. Therefore, why not use the reference vehicle's ground truth in their method?

**Questions:**

The proposed method provides pseudo-labels for the ego vehicle by directly passing the predicted results of the reference vehicle, which is a very naive approach. A semi-supervised learning framework is more flexible. What kind of performance effects would a semi-supervised learning framework have in this experimental setup? The ground truth of the reference vehicle is completely discarded in the subsequent method design, which is undoubtedly regrettable. Even if the center position of the ground truth has a deviation for the ego vehicle, the size of the bounding box is entirely correct, and this information is also very important. Please also respond to the comments in  Weaknesses.

**Details Of Ethics Concerns:**

Thank you for the author's response. However, my concerns have not been fully addressed, and the current version of the paper requires more experiments regarding the comparison of label-efficient methods, as well as comparative experiments with offboard methods, to validate the advancement of the method. I believe the authors will be able to further optimize these shortcomings in future versions. After careful consideration, I will raise my score to 5.

---

> ### Author Response · Authors · 2024-11-20
> **Response to Reviewer R3jy [1/2]**
>
> Thank you for the detailed feedback. After carefully reading it, we respectfully think there might be a misunderstanding, and we apologize if it has been caused by our paper organization. Below, we respond to most, if not all, of your comments at this stage. We apologize in advance if the response is long.
>
> ## Answer to Weakness 1: Clarification of our setting.
>
> We appreciate that you brought up the offboard setting and two references. After carefully reading them, we respectfully think that our paper does not belong to an offboard setting, and we explain it as follows.
>
> Our overall goal is to develop a 3D perception model that can be deployed in an online setting. A conventional development process can be decomposed into four stages typically.
> - Stage 1: data collection (online)
> - Stage 2: data annotation, often by humans (offline)
> - Stage 3: model training and validation (offline)
> - Stage 4: model deployment and evaluation - on a disjoint set of data (online)
>
> Our paper exactly follows the four stages, except that in Stage 1, we assume that some nearby agents (*e.g.*, robotaxi, roadside unit) share the predictions from their viewpoints as pseudo labels (*e.g.*, bounding boxes). Our main contribution is in how to leverage these pseudo labels to reduce human annotation in Stage 2 while maintaining the trained model’s quality in Stage 3. We will rewrite our problem setup to clarify this (Lines 157 - 161).
>
> **Onboard vs. offboard.**
>
> Despite such a difference, our Stage 4 strictly follows an online (onboard) setting. That is, one cannot rely on future frames (like the next several seconds) to enhance the current prediction. Our main result in Table 2 of the main paper exactly follows this setting.
>
> In our humble opinion, this is different from the offboard evaluation. If we understand the two papers correctly, their focus is on improving Stage 2. That is, suppose a portion of the training data is human-annotated, can we effectively use it to auto-label the remaining data? Since this Stage is offline, one can leverage future frames to improve the labeling of the current frame. For example, one can use data from -2:2 seconds to predict the labels at 0 seconds. Their main (offboard) evaluation is on how well the auto-labeling matches human annotation.
>
> **Potential cause of confusion.**
>
> We think Table 1 of the main paper might create the wrong impression that we aim for an offboard setting. We put Table 1 in front of Table 2 mainly to highlight the challenge of leveraging nearby agents’ pseudo labels (*i.e.*, initial boxes), but Table 1 is not our main result. We will clarify this in the final version.
>
> **Comparison to offboard setting.**
>
> That said, our paper and most of the label-efficient and domain adaptation papers (please see our related work) share a similar goal with the offboard setting: reducing human annotation effort (in Stage 2) for training the online perception model (in Stage 3).
>
> In the two offboard papers, they aim to use a human-labeled training data portion to auto-label the remaining portion. In our setting, we leverage the pseudo labels offered by nearby agents (Stage 1). We humbly think that these two directions are not competitors, but complementary to each other: using different initial signals to achieve a common goal. One can potentially combine them for better performance.
>
> We hope these explanations have resolved your concern. We will be more than happy to discuss further.

---

> > ### Author Response · Authors · 2024-11-20
> > **Response to Reviewer R3jy [2/2]**
> >
> > ## Answer to Weakness 2: About the contribution of the proposed method.
> >
> > **Q. … the proposed solution not only uses full data annotation to train an accurate detector …**
> >
> > Based on the four-stage process above, we respectfully think there is a misunderstanding.
> > We assume some nearby agents can offer pseudo labels based on their onboard perception models (in Stage 1). However, we are not in charge of building those perception models. We justify this setting in Lines 46 - 51 and 174 - 191, including how the existence of nearby agents is feasible. Once again, we think Table 1 might create the wrong impression as we include “*R’s GT*” there. We did so mainly to support our investigation in Section 3.2, especially Lines 201 - 205. We will clarify this in the final version.
> >
> > **Q. … However, the method itself is offline, which creates a contradiction. …**
> >
> > Likewise, based on the four-stage process above, we respectfully think there is a misunderstanding. We obtain nearby agents’ predictions online (on data portion A), and apply our method offline to train a 3D perception model (on data portion A). Once trained, we then test the model on a disjoint data portion B (*i.e.*, Table 2 of the main paper). We respectfully see no contradiction here.
> >
> > **Q. What are the proposed method's contributions in light of these concerns?**
> >
> > Our contribution, in a nutshell, is to propose and explore the new scenario—how to leverage other agents’ predictions collected online to facilitate offline model training. We systematically identify the challenges in Section 3.2, and accordingly develop solutions in Sections 3.3 and 3.4. The extra manual labeling is part of the solution in Section 3.3 (40 frames only, and can be replaced by synthetic data in Table 3 (a)), and we humbly do not think it diminishes our contribution but instead demonstrates the applicability of our method. It shows that a small amount of accurately annotated data can turn the noisy pseudo labels from nearby agents into effective training data for the 3D perception model. Please see the next response for more information, and we will be happy to clarify more.
> >
> > ## Answer to Weakness 3: Naive approach vs. semi-supervised learning.
> >
> > We want to emphasize that our proposed approach is built upon self-training (Section 3.4), which is indeed a popular approach in semi-supervised learning. Specifically, self-training uses the current pseudo labeled data (*e.g.*, from reference vehicles) to train a teacher model, which then generates better pseudo labels to re-train the model (student model). Please see Section 3.5, Steps 1 and 2, for this training procedure. Our key contribution is to show that even training the teacher model can be challenging due to mislocalization (Section 3.3) and viewpoint mismatch (Section 3.4). To this end, we propose an extremely label-efficient refinement (Section 3.3) and a curriculum-based self-training approach (Section 3.4).
> >
> > In sum, our proposed method is not as naive as directly passing the predicted results of the reference vehicle. Our experiments were also carefully designed to justify every proposed solution. For example, in Table 2 of the main paper, row 1 is the naive approach the reviewer mentioned, which does not involve self-training (hence, no semi-supervised learning). Rows 3 - 4 improve the predicted results, but still without self-training. Rows 5 -7 apply standard self-training (hence, standard semi-supervised learning). Rows 8-10 apply our proposed self-training, showing the best onboard detection accuracy, which is just a few percent shy of training a model with fully annotated data.
> >
> > ## Answer to Weakness 4: The ground truth of the reference vehicle is completely discarded in the subsequent method design, which is undoubtedly regrettable.
> >
> > We hope our responses above also answer your question. During online data collection (Stage 1), we obtain reference car’s predictions based on their onboard detectors. We do not use reference cars’ ground-truth labels because, in reality, we can never obtain them from reference cars unless they are equipped with human annotators who can label them in an online fashion. We include experiments using reference cars’ ground truths in Tables 1 and 2 only for analysis purposes. After all, the dataset (V2V4Real) we use is fully labeled, and we use it as a testbed to conduct our study (Lines 72 - 77).
> >
> > Again, we appreciate your valuable opinions, which will significantly improve our paper. Please also kindly give us a chance to clarify any remaining points that are unclear.

---

> > ### Comment · Reviewer_R3jy · 2024-11-22
> >
> > Thank you for the author's response. The offboard methods and the method in this paper are both based on a well-trained pre-trained model, so compared to the unsupervised baseline selected in Table 2, I still believe that the offboard methods are better baselines.  A more equitable experimental comparison is necessary for verifying the reliability of the paper's performance.
> >
> > Additionally, perhaps I am mistaken, but I believe that in the training of the ranker, this paper utilized manual annotations from the ego-vehicle view. This makes the comparison with the unsupervised method DRIFT in Table 2 even more unfair.

---

> ### Comment · Reviewer_R3jy · 2024-11-22
>
> Currently, many label-efficient methods use low annotation cost to correct pseudo-labels or to mine pseudo-labels. The authors should also discuss these methods:
>
> [1] Zhang D, Liang D, Zou Z, et al. A simple vision transformer for weakly semi-supervised 3d object detection[C]//Proceedings of the IEEE/CVF International Conference on Computer Vision. 2023: 8373-8383.
>
> [2] Liu C, Qian X, Huang B, et al. Multimodal transformer for automatic 3d annotation and object detection[C]//European Conference on Computer Vision. Cham: Springer Nature Switzerland, 2022: 657-673.
>
> [3] Xia Q, Deng J, Wen C, et al. Coin: Contrastive instance feature mining for outdoor 3d object detection with very limited annotations[C]//Proceedings of the IEEE/CVF International Conference on Computer Vision. 2023: 6254-6263.
>
> [4] Yang Y, Fan L, Zhang Z. Mixsup: Mixed-grained supervision for label-efficient lidar-based 3d object detection[J]. arXiv preprint arXiv:2401.16305, 2024.
>
> [5] Xia Q, Ye W, Wu H, et al. HINTED: Hard Instance Enhanced Detector with Mixed-Density Feature Fusion for Sparsely-Supervised 3D Object Detection[C]//Proceedings of the IEEE/CVF Conference on Computer Vision and Pattern Recognition. 2024: 15321-15330.
>
> [6] Liu C, Gao C, Liu F, et al. Ss3d: Sparsely-supervised 3d object detection from point cloud[C]//Proceedings of the IEEE/CVF conference on computer vision and pattern recognition. 2022: 8428-8437.
>
> In contrast to unsupervised methods, these approaches exhibit greater comparative value.

---

> ### Author Response · Authors · 2024-11-23
> **Thank you for your timely response [1/2]**
>
> Thank you so much for reading our long rebuttal and providing us with a timely response. If we understand correctly, our rebuttal has addressed many of your concerns that resulted from a different understanding of our paper. In the following, we would like to provide our response to each of your new concerns.
>
> **“The offboard methods and the method in this paper are both based on a well-trained pre-trained model, …, I still believe that the offboard methods are better baselines.”**
>
> **“ … so compared to the unsupervised baseline selected in Table 2, … A more equitable experimental comparison is necessary …”**
>
> Thank you for your further comment, which allows us to better address your concern.
>
> We agree that on the surface, both the offboard papers and our paper assume the existence of a pre-trained model. However, the accessibility to the model is drastically different, creating two distinct settings that are not directly comparable.
>
> More specifically, suppose the overall goal is to develop an accurate onboard detector for the ego car. In the offboard papers, the pre-trained offboard model is directly accessible. One can use it to label the unlabeled data collected by the ego car ($X_E$ in Line 159). The resulting pseudo-labeled data can then be used to train the final onboard model.
>
> However, in our paper, we do not have direct access to the pre-trained model, as it is deployed on the nearby agent (thus denoted by $f_R$ in Line 158), not the ego car. As such, we cannot use it to label the unlabeled data collected by the ego car. What we can access are the nearby agent’s predictions on the data it collects ($X_R$ in Line 159), and we attempt to use them as pseudo labels of $X_E$ to train the ego car’s onboard model. (Please see Lines 46 - 51 and Lines 176 - 191 for the feasibility and practicality of our setting.)
>
> We note that the predictions on the nearby agent’s data ($f_R(X_R)$ in Line 161) are quite different from the predictions on the ego car’s data ($f_R(X_E)$), even if we represent them in the global coordinate. This is evidenced in Table 1, where the 5th-row “sharing detector” means one can access the pre-trained model to label the ego car’s data directly; the 1st-row means one can only access $f_R(X_R)$. The huge gap in pseudo label quality between the two rows (22.7% on Recall and 38.8% on Precision) highlights the core problem we aim to address, which results from mislocalization and viewpoint mismatch (please see Lines 205 - 215).
>
> We also want to note that the unsupervised baseline in Table 2 can also receive $f_R(X_R)$ like ours. In other words, they have the equitable, same “limited” access to the pre-trained detector deployed on the nearby agent.
>
> In sum, we appreciate the reviewer’s comment. However, we respectfully think the settings in the offboard papers and ours are quite different and not directly comparable. As such, we respectfully disagree that the offboard papers are better baselines: our setting does not allow direct access to the pre-trained model, and nearby agents cannot predict labels in an offboard fashion (as they need to make online predictions to drive safely). In our humble opinion, it makes little sense to claim which setting is “better,” as they stand for different scenarios where one can obtain supervision. In particular, our paper investigates a new problem — can we leverage nearby agents’ predictions (on the data they perceive) to train an accurate model for the ego agent?

---

> > ### Author Response · Authors · 2024-11-23
> > **Thank you for your timely response [2/2]**
> >
> > **“Additionally, perhaps I am mistaken, but I believe that in the training of the ranker, this paper utilized manual annotations from the ego-vehicle view. This makes the comparison with the unsupervised method DRIFT in Table 2 even more unfair.”**
> >
> > You are right, training the ranker requires labeled data on the ego car’s data. In Table 2, we mainly reported the results using a handful of human-annotated data, and we explored using simulated data (no human annotation needed) in Table 3 (a) (*cf.* Lines 90 - 91 and Lines 468 - 470). As shown, a ranker trained with off-the-shelf simulated data can achieve 28.4% AP at IoU 0.7, higher than the 11.0% AP at IoU 0.7 achieved by DRIFT (Table 2 row 9). We appreciate your valuable comment and will include this result in Table 2 and clearly indicate in Table 2 which rows require human annotations.
> >
> > Besides the above comparison, we want to reiterate that our paper is not merely about pseudo label correction. Instead, we hope our response above has clarified that we study a novel, different problem setting from the two offboard papers. In our humble opinion, our main contributions are proposing a new learning scenario (*i.e.*, leveraging nearby agents’ predictions), systematically analyzing its challenges, and presenting the very first solution (*i.e.*, R&B-POP, where label-efficient ranker training is a component, not the whole picture), which has been recognized by the other reviewers (iYjB, LF3v, goE2, dJeW).
> >
> > As the very first paper to investigate a new scenario, we respectfully think our main mission is not to claim “our solution is better than prior ones,” but to point out the inherent challenges and explore “directions” to resolve them so that future work has “reference” and “baselines” to build upon.
> >
> > With such a mindset, we certainly do not view our proposed ranker as a “better approach” than the unsupervised baseline (*i.e.*, DRIFT), but as a “practical solution” to address what it struggles with. As discussed in Lines 227 - 232, the mislocalization between the nearby and ego car is not subtle but substantial and must be addressed. We include DRIFT in Table 2 mainly to show that the problem cannot be easily addressed by a purely unsupervised approach, but can be largely mitigated with the aid of a ranker trained supervisedly with either a handful of human-annotated data or simulated data. We will certainly clarify this in our final paper version.
> >
> > **“Currently, many label-efficient methods use low annotation cost to correct pseudo-labels or to mine pseudo-labels. The authors should also discuss these methods:”**
> >
> > Thank you for providing these references. We will surely cite them and discuss them in the related work (*e.g.*, in the Label-efficient learning paragraphs).
> > We also appreciate your comments and concerns about our ranker because it was trained on a few labeled data and shares similarities with label-efficient methods. We want to emphasize that our study does not aim to beat existing label-efficient methods. Rather, we want to reinforce the importance of pseudo label correction and selection and how they can be achieved label-efficiently and effectively if incorporated with problem-specific characteristics.
> > For instance, we found that the poor pseudo label qualities provided by the nearby agents mainly resulted from mislocalization (GPS and synchronization errors) and viewpoint mismatch, and introduced tailored approaches to correct and select high-quality pseudo labels. Such causes are specific to our new scenario and not commonly seen in existing works whose poor pseudo labels mainly result from models trained with insufficient data.
> >
> > In light of our additional explanation and clarification, we hope that we have addressed most, if not all, of your concerns. If so, we would appreciate it if the reviewer would reconsider the rating. Thank you.

---

> > > ### Author Response · Authors · 2024-11-27
> > >
> > > We thank the reviewer again for your timely response and invaluable comments. Based on your feedback, we have revised our manuscript. Please don’t hesitate to let us know if you have any remaining concerns. Additionally, **we are currently conducting experiments to compare with a semi-supervised method**, as suggested. We anticipate sharing the results soon and sincerely appreciate your patience!

---

> > > > ### Author Response · Authors · 2024-12-03
> > > >
> > > > Dear Reviewer R3jy,
> > > >
> > > > We would like to give you some updates:
> > > > - First, due to some computational issues, we need to delay a little bit about semi-supervised learning experiments, but we will surely try our best to provide the result by the end of the discussion period.
> > > > - Second, once again, we thank you for all the valuable comments and active involvement in the author-review discussion. We want to remind you that the author-reviewer discussion will end tomorrow. While we respectfully think that we have appropriately addressed your concerns, we will be willing to answer any remaining questions you have. Please kindly provide us with the feedback you want us to discuss further by December 3rd so that you may consider increasing your score.

---

> > > > > ### Author Response · Authors · 2024-12-04
> > > > >
> > > > > Dear R3jy,
> > > > >
> > > > > We appreciate your increasing the rating and your further comments on the updated official review form. (We almost overlooked it as we did not receive an email notification for such an update made on Dec 2.)
> > > > >
> > > > > While you put the comments in the “Details Of Ethics Concerns” field, we respectfully believe the comments are irrelevant to ethical issues, but are suggestions to further strengthen our submission.
> > > > >
> > > > > Your suggestions are well received and we will be happy to include additional comparisons to label-efficient methods and offboard methods.
> > > > >
> > > > > Meanwhile, we want to note that we have added an experiment regarding direct semi-supervised learning in the general response. The experimental results demonstrate two things. First, reference cars’ predictions are valuable auxiliary supervision. Second, such information and our method are compatible with direct semi-supervised learning to further boost performance. We respectfully believe these results validate the value and advancement of our method.
> > > > >
> > > > > In our humble opinion, offboard methods, other semi-supervised (or label-efficient) algorithms, and our method leverage different/orthogonal information for a common goal—how to learn perception without extensive human labeling. For instance, label-efficient methods investigated how to use limited labels; offboard methods investigated how to use future-frame information (in an offboard setting); and ours investigated a novel scenario using nearby agents’ predictions as supervision to learn perception. As such, these methods are not competitors but complementary if the corresponding information is available, as demonstrated in our additional experiment.
> > > > >
> > > > > While we cannot explore this complementariness further due to the limited response period to your further comment, we will be willing to investigate more in the final version.
> > > > >
> > > > > Once again, we appreciate that you raised the score, and we are happy to know that our responses have addressed your major concerns, especially the clarification in our setting.

---

### Author Response · Authors · 2024-11-20
**General response**

We thank all reviewers for their valuable comments. We particularly appreciate that the reviewers recognized that our proposed scenario is interesting, novel, and creative (iYjB, goE2, dJeW); our method is well-motivated, well-explained, and intuitive (R3jy, LF3v, goE2, dJeW); our experiments and analyses are thorough and well-designed (R3jy, iYjB, LF3v, goE2, dJeW). We humbly think that these are essential for introducing a new scenario that the community can follow and further improve. Below, we have posted our initial responses to each reviewer to address most of the concerns, if not all. We would appreciate the reviewers’ timely feedback so we can further discuss it.

---

### Author Response · Authors · 2024-11-27

We sincerely appreciate the reviewers’ thoughtful and constructive feedback. Based on your suggestions, we have made improvements to our manuscript. The updated text has been highlighted in red. A summary of the changes is as follows:

- Added new references on offboard and semi-supervised methods in Related Work. (**R3jy**)
- Included a discussion on offboard methods in Section S1. (**R3jy**)
- Expanded the problem definition and setup in Section S1 to provide more clarity. (**R3jy**, **LF3v**)
- Justified the ranker design in Section S2 under the paragraph on ranker architecture. (**LF3v**)
- Provided additional explanation about the experiment with the ranker without any human labels in Lines 469 - 470. (**R3jy**, **goE2**)
- Added experimental results (Table S1) describing the impact of training data size on detector performance in Section S3.1. (**iYjB**)

We are also conducting further experiments to compare our approach with a semi-supervised method, as suggested by the reviewers. **These additional results are in progress, and we anticipate sharing them soon.** We appreciate your understanding and patience in waiting for this analysis.

Thank you again for your valuable feedback, which has greatly strengthened our work!

---

### Author Response · Authors · 2024-12-04
**Additional general response [1/2]**

Dear reviewers,

We appreciate all the reviewers’ review comments and interactions during the discussion period. We provide further general responses as follows.

### Comparison to label-efficient, semi-supervised learning methods

We apologize for the delay in providing the requested results investigating other semi-supervised learning methods (R3jy, goE2, dJeW). In the past week, we suffered an unexpected server shutdown. Fortunately, we managed to have the results ready, as follows.

To begin with, let us reiterate our main contribution, which is to explore a novel scenario of using the reference car’s predictions as supervision to train the ego car’s detector. In our main results, we annotated 40 frames of the ego car’s data to train the ranker to refine the reference car’s predictions (Line 286). The refined predictions were then used as pseudo labels to train the object detector for the ego car (Section 3.3). We note that our approach already involves semi-supervised learning, using self-training to fill in missing labels and, in turn, re-train the model (Section 3.4). **We have shown that it is feasible to bypass the 40 annotated frames but use synthetic data to train the ranker** (Table 3a and Lines 469 - 470; see also responses to R3jy).

As reviewers R3jy, goE2, and dJeW suggested, we have investigated a direct semi-supervised approach, using the 40 annotated frames and other unannotated frames of the ego car’s data to train the detector. We apply 3DIoUMatch [a], a widely used and representative semi-supervised learning approach in this setting. We note that the official code of 3DIoUMatch used the PV-RCNN detector, not the PointPillar detector in our main paper. As such, we rerun our approach using the PV-RCNN detector for a fair comparison.

*Table: Additional experiments on semi-supervised approach. The performance is reported on AP at IoU 0.7. * and $\dagger$ indicate supervision and approach, respectively.*
| labeled 40 frames$^{*}$ | _R_'s pred$^{*}$ | 3DIoUMatch$^{\dagger}$ [a] | Ranker$^{\dagger}$ (Sec 3.3) | Curriculum$^{\dagger}$ (Sec 3.4) | 0-30m | 30-50m | 50-80m | 0-80m |
|:-----------------:|:--------:|:--------------:|:----------------:|:--------------------:|:-----:|:------:|:------:|:-----:|
|         o         |          |        o       |                  |                      |  60.7 |  24.4  |   6.0  |  37.1 |
|         o         |     o    |                |         o        |           o          |  65.1 |  34.1  |  12.6  |  41.7 |
|         o         |     o    |        o       |         o        |                      |  60.6 |  30.7  |   8.6  |  39.5 |
|         o         |     o    |        o       |         o        |           o          |  68.1 |  36.8  |  12.9  |  44.4 |


The above table summarizes the results. Row 1 is the result of 3DIoUMatch [a], and Row 2 is the result of our approach (cf. Row 10 in Table 2, but using PV-RCNN as the detector). We see that our approach outperforms 3DIoUMatch, demonstrating the value of using reference cars’ predictions as auxiliary supervisions.

That said, we certainly do not claim that our approach is a better semi-supervised learning approach than 3DIoUMatch (or other direct semi-supervised methods). First, we leverage reference cars’ predictions. Second, ours and 3DIoUMatch use the 40 annotated frames in drastically different ways—ours uses them for ranker training, while 3DIoUMatch uses them for detector training. As such, they are potentially complementary to each other. If integrated appropriately, they may lead to even better results.

To investigate this idea, we use our ranker to refine reference cars’ predictions and add those high-quality ones (_i.e._, < 40 meters, defined in Section 3.4) as extra labels to 3DIoUMatch. Row 3 shows the results: we see a 2.4% boost in 0-80 meters against Row 1, justifying the compatibility of ours and 3DIoUMatch. On top of Row 3, we further apply our distance-based curriculum for self-training (Section 3.4), using 3DIoUMatch’s predictions on all the data as pseudo labels to re-train the detector. Row 4 shows the results: we see another 4.9% boost against Row 3 and 2.7% boost against Row 2. In sum, these results demonstrate 1) the effectiveness of our approach in leveraging reference cars’ predictions as supervision (Row 3 and Row 2 vs. Row 1) and 2) the compatibility of our approach with existing direct semi-supervised learning approaches to further boost the accuracy (Row 4 vs. Row 3 and Row 2). We view such compatibility as a notable strength: it demonstrates our approach as a valuable add-on when nearby agents’ predictions are available.

Once again, we appreciate the reviewers pointing out an important topic to discuss. We will include these findings in our final version.

---
[a] Wang et al., "3dioumatch: Leveraging iou prediction for semi-supervised 3d object detection," in CVPR 2021.

---

> ### Author Response · Authors · 2024-12-04
> **Additional general response [2/2]**
>
> ### Discussion on inherent diversity of approaches for self-driving
>
> We thank the reviewers for mentioning offboard methods and other semi-supervised (or label-efficient) algorithms. All these methods (including ours) have a common goal—how to learn perception without extensive human labeling. However, instead of sticking with a fixed setting, they explore different settings/scenarios, leveraging different and orthogonal information for the goal. For instance, label-efficient methods investigated how to use limited labels; offboard methods investigated how to use future-frame information (in an offboard setting); and ours investigated a novel scenario using nearby agents’ predictions as supervision to learn perception. (Some other methods like [b] use data from repeated routes, if accessible.) In our humble opinion, these methods are NOT competitors BUT complementary if the corresponding information is available, as demonstrated in our response above.
> We have strengthened the discussion in this aspect in the related work and will add more in the final version.
>
> Overall, we think what makes autonomous driving an interesting research area is not only its potential impact on society but also the diversity of data cues to leverage. We humbly think exploring different data cues is a valuable direction. Individually, they have their appropriate application scenarios. Collectively, they may complement each other to further boost the perception accuracy and robustness. We will include this in the discussion in the final version.
>
> ---
> [b] You et al., “Learning to detect mobile objects from lidar scans without labels,” in CVPR 2022.

---

### Meta-Review · Area_Chair_LuWv · 2024-12-20

**Metareview:**

This paper received mixed reviews, with four reviewers voting for borderline acceptance and one reviewer (R3jy) voting for borderline rejection. The key contributions of this paper include the introduction of a new label-efficient setting for 3D object detection, which leverages prediction results from nearby agents, and the design of the first solution to address this novel setting.

After the rebuttal, Reviewers LF3v, goE2, and dJeW acknowledged that most of their concerns had been addressed, while Reviewer iYjB did not respond, and Reviewer R3jy actively engaged in the discussion with the authors. Since Reviewer iYjB also did not participate in the AC-reviewer discussion period, the AC reviewed the authors' response to their questions and believes that most concerns were resolved.

The remaining concerns from reviewers focused on insufficient comparisons with label-efficient methods (R3jy, dJeW) and offboard methods (R3jy). The AC noted that neither Reviewer R3jy nor dJeW responded to the authors' latest reply. After carefully reviewing the authors' follow-up responses, the AC agrees with the authors' argument that offboard 3D object detection does not constitute a directly comparable setting to the proposed method, even though both aim to generate pseudo labels from existing models. Regarding the comparison with other label-efficient methods, such as semi-supervised methods, the authors addressed this concern by providing a comparison with 3DIoUMatch and demonstrating a performance improvement through the incorporation of a semi-supervised technique.

As a result, the AC believes that the concerns raised by Reviewer R3jy have largely been addressed. Therefore, the AC recommends accepting the paper and encourages the authors to include the newly added experimental results in the final version.

**Additional Comments On Reviewer Discussion:**

This paper received mixed reviews, with four reviewers voting for borderline acceptance and one reviewer (R3jy) voting for borderline rejection. The key contributions of this paper include the introduction of a new label-efficient setting for 3D object detection, which leverages prediction results from nearby agents, and the design of the first solution to address this novel setting.

After the rebuttal, Reviewers LF3v, goE2, and dJeW acknowledged that most of their concerns had been addressed, while Reviewer iYjB did not respond, and Reviewer R3jy actively engaged in the discussion with the authors. Since Reviewer iYjB also did not participate in the AC-reviewer discussion period, the AC reviewed the authors' response to their questions and believes that most concerns were resolved.

The remaining concerns from reviewers focused on insufficient comparisons with label-efficient methods (R3jy, dJeW) and offboard methods (R3jy). The AC noted that neither Reviewer R3jy nor dJeW responded to the authors' latest reply. After carefully reviewing the authors' follow-up responses, the AC agrees with the authors' argument that offboard 3D object detection does not constitute a directly comparable setting to the proposed method, even though both aim to generate pseudo labels from existing models. Regarding the comparison with other label-efficient methods, such as semi-supervised methods, the authors addressed this concern by providing a comparison with 3DIoUMatch and demonstrating a performance improvement through the incorporation of a semi-supervised technique.

As a result, the AC believes that the concerns raised by Reviewer R3jy have largely been addressed. Therefore, the AC recommends accepting the paper and encourages the authors to include the newly added experimental results in the final version.

---

### Decision · Program_Chairs · 2025-01-22

Accept (Poster)